# Teacher-rated aggression and co-occurring behaviors and emotional problems among schoolchildren in four population-based European cohorts

Alyce M. Whipp[1]*, Eero Vuoksimaa[1], Koen Bolhuis[2], Eveline L. de Zeeuw[3], Tellervo Korhonen[1], Matteo Mauri[4], Lea Pulkkinen[5], Kaili Rimfeld[6], Richard J. Rose[7], Catharina (Toos) E. M. van Beijsterveldt[3], Meike Bartels[3,8], Robert Plomin[6], Henning Tiemeier[9], Jaakko Kaprio[1,10], Dorret I. Boomsma [3,8]

1 Institute for Molecular Medicine Finland, HiLIFE, University of Helsinki, Helsinki, Finland, 2 Department of Child and Adolescent Psychiatry, Erasmus Medical Center, Rotterdam, The Netherlands, 3 Department of Biological Psychology, Faculty of Behavioural and Movement Sciences, Vrije Universiteit Amsterdam, Amsterdam, The Netherlands, 4 University of Cagliari, Cagliari, Italy, 5 Department of Psychology, University of Jyvaskyla, Jyvaskyla, Finland, 6 Social, Genetic, and Developmental Psychiatry Centre, Institute of Psychiatry, Psychology, and Neuroscience, King's College London, London, United Kingdom, 7 Department of Psychological & Brain Sciences, Indiana University, Bloomington, Indiana, United States of America, 8 Amsterdam Public Health Research Institute, Amsterdam, The Netherlands, 9 Department of Social and Behavioral Science, Harvard TH Chan School of Public Health, Boston, Massachusetts, United States of America, 10 Department of Public Health, University of Helsinki, Helsinki, Finland

* alyce.whipp@helsinki.fi

**Data Availability Statement:** The data that support the findings of this study are not publicly available due to privacy or ethical restrictions. FinnTwin12

## Abstract

Aggressive behavior in school is an ongoing concern. The current focus is on specific manifestations such as bullying, but the behavior is broad and heterogenous. Children spend a substantial amount of time in school, but their behaviors in the school setting tend to be less well characterized than at home. Because aggression may index multiple behavioral problems, we used three validated instruments to assess means, correlations and gender differences of teacher-rated aggressive behavior with co-occurring externalizing/internalizing problems and social behavior in 39,936 schoolchildren aged 7–14 from 4 population-based cohorts from Finland, the Netherlands, and the UK. Correlations of aggressive behavior were high with all other externalizing problems ($r$: 0.47–0.80) and lower with internalizing problems ($r$: 0.02–0.39). A negative association was observed with prosocial behavior ($r$: -0.33 to -0.54). Mean levels of aggressive behavior differed significantly by gender. Despite the higher mean levels of aggressive behavior in boys, the correlations were notably similar for boys and girls (e.g., aggressive-hyperactivity correlations: 0.51–0.75 boys, 0.47–0.70 girls) and did not vary greatly with respect to age, instrument or cohort. Thus, teacher-rated aggressive behavior rarely occurs in isolation; boys and girls with problems of aggressive behavior likely require help with other behavioral and emotional problems. Important to note, higher aggressive behavior is not only associated with higher amounts of other externalizing and internalizing problems but also with lower levels of prosocial behavior.

data are available through the Institute for Molecular Medicine Finland (FIMM) Data Access Committee (DAC) for authorized researchers who have IRB/ethics approval and an institutionally approved study plan. For more details, please contact the FIMM DAC (fimm-dac@helsinki.fi). The Generation R datasets analyzed during the current study are not publicly available due to the terms and conditions participants agree to when they participate in Generation R, but are available from data management on reasonable request (datamanagementgenr@erasmusmc.nl). Applications for data and information on the access policy for the Netherlands Twin Register may be found at: https://tweelingenregister.vu.nl/submitting-a-data-sharing-request. The Twins Early Development Study's data access policy can be found at the following link https://www.teds.ac.uk/researchers/teds-data-access-policy.

**Funding:** This work is part of the ACTION consortium which is supported by funding from the European Union Seventh Framework Programme (FP7/2007-2013) under grant agreement no. 602768. FT12: Data collection has been supported by the National Institute of Alcohol Abuse and Alcoholism (grants AA-12502, AA-00145, and AA-09203 to RJR) and the Academy of Finland (grants 100499, 205585, 118555, 141054 and 264146 to JK). JK has been supported by the Academy of Finland (grants 308248, 312073). GENR: This work was supported by the Netherlands Organization for Scientific Research (NWO-grant 016.VICI.170.200) to HT, and the Sophia Children's Hospital Research Foundation (research fellowship grant 921) to KB. Super computing resources were made possible through the NWO Physical Sciences Division (surfsara.nl). The first phase of the Generation R Study is made possible by financial support from the Erasmus Medical Centre, Rotterdam; the Erasmus University Rotterdam; The Netherlands Organization for Health Research and Development (ZonMw). NTR: Data collection in the NTR was funded by the Netherlands Organization for Science (NWO): Twin-family database for behavior genetics and genomics studies (NWO-480-04-004); Gravitation program of the Dutch Ministry of Education, Culture, and Science and the Netherlands Organization for Scientific Research (NWO-024.001.003). 'Longitudinal data collection from teachers of Dutch twins and their siblings' (NWO-481-08-011); 'Twin-family study of individual differences in school achievement' (NWO-056-32-010); ZonMW "Genetic influences on stability and change in psychopathology from childhood to young adulthood" (NWO-912-10-020); "Netherlands Twin Registry Repository" (NWO-

## Introduction

Aggressive behavior in school is a persistent topic of concern, with bullying, gender- and sexuality-based misconduct, and extreme violence (e.g., school fights, stabbings, shootings) currently garnering the most public attention [1, 2]. While these specific manifestations of aggression are important to understand and manage for the safety of all in the school environment, broadly defined aggression may be an important marker for a wider set of behavioral problems that co-occur with it. Aggressive behavior itself is heterogenous in nature and can involve overt and covert as well as planned and unplanned actions and rarely occurs in isolation [3–5]. All children express such behaviors to varying degrees across development. Among school-aged children, aggressive behavior is associated with relationship quality among their peers, teachers, and families, as well as their eductional attainment, risk for substance abuse and other psychiatric disorders, and risk for partipation in criminal activities [6–8].

Aggressive behavior, in general, is often characterized and studied via parental or self ratings, with teacher ratings less often collected in epidemiologic or clinical research. Research from the past several decades has shown that an individual's aggressive behavior is often situational (e.g., present at school, but not at home) and that assessments of behavioral and emotional problems do not correlate well between raters [9–12]. In the school setting, teacher, peer, and self ratings have been used to characterize behavioral problems; peer and teacher ratings are more strongly correlated compared to other observer correlations [9, 13], and self ratings are commonly used in later adolescence. Teacher-rated measures of behavioral problems have also shown high reliability [13–15]. Furthermore, studies have shown that teacher ratings of problem behaviors have often been more useful than parent ratings in diagnostic and predictive outcomes, including psychiatric disorders and criminality [7, 10, 16, 17]. Teachers' ability to observe children in a structured setting and among peers of similar ages and abilities provides them with a valuable comparison-base, making their insight important.

In investigating teacher-rated aggressive behavior, we also need to consider co-occuring behaviors and emotional problems. Aggressive behavior often co-occurs with other externalizing behaviors (e.g., hyperactivity) [5, 18–20]. Internalizing problems, such as depression or anxiety, are also shown to moderately co-occur with aggressive behavior [5, 21–24]. Furthermore, children with aggression and co-morbidities often have more than one co-morbidity, and, compared to aggressive children without co-morbidities, have poorer outcomes [8, 18, 25, 26]. Lastly, aggressive behavior has also been shown to be negatively associated with prosocial behavior [27, 28]. Once thought to be opposite ends of the spectrum [27], prosocial behavior now appears to have a more complex relationship with aggressive behavior, with suggestions that prosocial behavior can be protective against future negative outcomes in aggressive children [29] and that there is possibly a goal-oriented prosocial behavior subtype positively associated with aggressive behavior [30].

In studying aggressive behavior, as well as related externalizing, internalizing and prosocial behavior, gender differences have played a large and complex role in our understanding. Early research on aggressive behavior focused only or mainly on boys, although studies in recent decades have greatly enriched our knowledge regarding both genders. Distinctions between direct aggression as being higher among boys and indirect aggression being higher among girls have generally given way to an understanding that both genders engage in both types of aggression, and that at least for indirect aggression the gender differences, if any, are small [31, 32]. Furthermore, a study by Nivette et al. [33] indicates that in societies that experience high gender inequality, differences in physical aggression by gender are minimal. In addition to mean differences, it is also important to investigate if behavioral problems and prosocial behavior are similarly or differently related to aggression in boys and girls.

480-15-001/674). DIB would like to acknowledge the KNAW Academy Professor Award (PAH/6635). TEDS: TEDS is supported by a program grant to RP from the UK Medical Research Council (MR/M021475/1). No funders had any involvement in the conduct of this research, including in the study design, data collection, analysis and interpretation of the data, the writing of the report, and the decision to submit to the article for publication.

**Competing interests:** All authors declare no conflicts of interest.

Regarding the school setting, research has often focused on bullying or psychiatric aggression diagnoses (e.g., conduct disorder, oppostional-defiant disorder) [19, 22, 34, 35], with less attention on general, non-pathological levels of aggressive behavior and co-occuring behaviors. To shed light on the phenotype of aggressive behavior in schools and how common co-occuring behaviors generally are, we carried out a unique study in four large population-based cohorts that assembled teacher ratings of schoolchildren by using three validated teacher-rating instruments.

This study's overall objective was to characterize the levels and associations of teacher-rated aggressive behavior with co-occurring behaviors of children aged 7–14. Large datasets from collaborating population-based cohorts of children from Finland, the Netherlands, and the UK were analyzed. Specific aims were: 1) to report the mean levels of aggressive behavior and other behaviors as well as emotional problems as assessed by teachers, 2) to examine associations (co-occurrence) of aggressive behavior with other behaviors and emotional problems, and 3) to assess gender differences in the mean levels and associations of aggressive behavior and co-occurring behaviors and emotional problems.

## Materials and methods

The datasets for this investigation were obtained through the collaboration of the ACTION (Aggression in Children: Unravelling gene-environment interplay to inform Treatment and InterventiON strategies; http://www.action-euproject.eu/) consortium [5]. Of the seven large child/adolescent cohorts brought together in ACTION, four had collected teacher rating data. In total, the data included 39,936 teacher ratings on children at ages 7, 9, 10, 12, and 14 (48.9% boys) from Finland, the Netherlands, and the UK, with some children observed at two or more ages. As in Bartels et al. [5], in the cohorts that included twin data one twin per family was randomly selected for analysis. Individuals were excluded if they had an illness or disability that interfered with their daily functioning (e.g., Down syndrome or severe neurodevelopmental disorder), and if they were missing substantial data from one of the examined subscales from cohort-respective behavioral and emotional problem questionnaires (see also: http://www.action-euproject.eu/content/data-protocols). Brief cohort and behavioral questionnaire descriptions are presented here (see S1 Text for further details on teacher rating collections and the school systems in Finland, the Netherlands, and the UK).

### Cohort descriptions

The FinnTwin12 (FT12) dataset was established in Finland from the cohort of twins born 1983–87 [36]. It is a population-based twin study aimed at examining health-related behaviors and their precursors. Data were collected at ages 12, 14, 17, and 22. Teacher ratings on behavioral and emotional problems were collected at ages 12 and 14. Response rates were 93% and 94% for age 12 and 14 teacher ratings, respectively.

The Generation R (GENR) study is a population-based birth cohort, established in the Netherlands from children born 2002–06 in the city of Rotterdam and surrounding areas [37], aimed at examining growth, development and health from fetal life to young adulthood. Data were collected during pregnancy, at birth, and frequently throughout childhood (currently up to age 13). Teacher ratings on behavioral and emotional problems were collected at child mean age 7 years; the response rate was 77%.

The Netherlands Twin Register (NTR) was established in the Netherlands in 1987 and remains ongoing. The register includes adults and young twins who were registered by their parents shortly after birth and who are followed longitudinally to examine, in particular, behavioral development and psychopathology [38]. Data are collected at ages 1, 2, 3, 5, 7, 9, 10, 12, 14, 16, and 18 years (at which point twins move into the adult twin register). In school year

1999–2000, the NTR began collecting teacher ratings for 7, 10 and 12 year old twins; the average response rate was ~60%.

The Twins Early Development Study (TEDS) was established in the UK from the cohort of twins born 1994–96 [39]. It is a UK-representative longitudinal twin study aimed at examining language, cognitive, and behavioral development. Data were collected at ages 2, 3, 4, 7, 8, 9, 10, 12, 14, 16, 18, and 21. Teacher ratings on behavioral and emotional problems were collected at ages 7, 9, and 12; response rates were 85%, 76%, 78%, respectively.

## Ethical statement

All data in the current investigation were collected under protocols that have been approved by the appropriate ethics committees, and studies were performed in accordance with the ethical standards established in the 1964 Declaration of Helsinki and its later amendments. For FT12, ethics approval was obtained from the Helsinki University Hospital Ethics Committee (HUS/845/2017) and the Indiana University Bloomington IRB (IRB-IUB, IRB00000222).

For GENR, study protocols were approved by the Medical Ethics Committee of the Erasmus Medical Center (NL55105.078.15). For NTR, ethics approval came from the Central Ethics Committee on Research Involving Human Subjects of the VU University Medical Center, Amsterdam, an Institutional Review Board (IRB) certified by the U.S. Office of Human Research Protections (IRB00002991 under Federal-wide Assurance FWA00017598; IRB/institute codes, NTR 03–180). For TEDS, ethical approval was provided by the King's College London ethics committee (reference number: PNM/09/10-104). All participants (or their guardians) provided written informed consent before participation in their respective cohorts.

## Study questionnaires

The Multidimensional Peer Nomination Inventory (MPNI), used by the FT12 cohort, was originally developed as a tool for rating peers on childhood social behavior. However, it has been adapted and modified to collect ratings from other raters, including teachers [13]. It is a 37-item questionnaire with 6 of the subscales used here: aggression (6 items), depression (5 items), hyperactivity–impulsivity (7 items), inattention (4 items), prosocial behavior (12 items), and social anxiety (2 items). The aggression scale includes both direct (e.g., calls people names when angry at them) and indirect (e.g., spreads rumors) aggression items. Although further subscale scoring was not performed, the MPNI internal structure based on factor analysis indicates aggression, hyperactivity–impulsivity and inattention subscales as part of a behavioral problem/externalizing subscale, and depression and social anxiety subscales as part of an emotional problems/internalizing subscale. Each item on the teacher rating questionnaire has four response choices (from 'not observed in child' to 'clearly observed'). Response choices are scored 0–3, and subscales are formed by taking the mean of all items in the subscale (no missing values were allowed). The internal consistency (Cronbach's alpha) values of the MPNI teacher ratings range from 0.69 (social anxiety scale among girls) to 0.94 (hyperactivity-impulsivity and externalizing behavior for boys), indicating high reliability [13].

The Strengths and Difficulties Questionnaire (SDQ), used by the TEDS cohort, is a 25-item questionnaire that measures common childhood mental health problems [40]. The five subscales of the SDQ are conduct problems (5 items), emotional problems (5 items), hyperactivity (5 items), peer problems (5 items), and prosocial (5 items). The SDQ recognizes conduct problems and hyperactivity subscales as part of an externalizing subscale, and emotional and peer problems subscales as part of an internalizing subscale. For this study, we use 'conduct problems' as a proxy for aggressive behavior, and 'emotional problems' as a proxy for anxiety problems, as in Bartels et al. [5]. The conduct problems scale includes both aggression items (e.g.,

often fights with other children) and rule-breaking type of items (e.g., often lies or cheats). Each item on the questionnaire has three response choices: 'not true', 'somewhat true', and 'certainly true'. Response choices range from 0–2 and are coded so that higher scores represent greater risk for the behavior attribute, and subscale scores are derived as a scaled mean score when at least 3 of the 5 items in a subscale are non-missing. The internal consistency (Cronbach's alpha) values of the SDQ teacher ratings range from 0.70 (peer problems) to 0.88 (hyperactivity-inattention), indicating high reliability [41].

The Teacher Report Form (TRF), part of the Achenbach System of Empirical-Based Assessment's (ASEBA), was collected by the GENR and NTR cohorts and is a 112-item questionnaire that measures childhood behavioral and emotional problems [42]. The 8 syndrome subscales used here are aggressive behavior (20 items), anxious/depressed (16 items), attention problems (26 items), rule-breaking behavior (12 items), social problems (11 items), somatic complaints (9 items), thought problems (1 item), and withdrawn/depressed (8 items). The aggressive behavior scale includes, for example, both direct actions (e.g., physically attacks people) and broader behaviors (e.g., easily frustrated). The TRF recognizes aggressive behavior and rule-breaking behavior as externalizing problems, and anxious/depressed, withdrawn/depressed and somatic complaints as internalizing problems. In this study, the attention problems subscale will also be grouped under externalizing problems. Each item on the questionnaire has three response choices: 'not true', 'somewhat or sometimes true', and 'very true or often true'. Response choices are scored 0–2, and items are summed to created sum scores of individual subscales. A small number of missing values per subscale (depending on the total number of items per subscale) was allowed by replacing the missing values with the mean item score of a subscale. The mean internal consistency (Cronbach's alpha) values of the TRF teacher ratings used in 21 countries range from 0.65 (thought problems) to 0.93 (attention problems), indicating high reliability [43].

## Analyses

Data from the FT12, NTR, and TEDS cohorts were analyzed at one site (by AMW) in Stata version 15 (Stata Corporation, College Station, TX, USA), while data from GENR were analyzed locally (by KB) using R version 3.3.2. The analyses followed a standard operating procedure [5] to ensure uniform data handling and analysis.

First, means, standard deviations (SD) and standard errors (SE) were obtained for all subscales for each age and cohort, separately by gender. T-tests were performed to determine if gender differences existed (p-value <0.05 was considered significant), and effect sizes were calculated as Cohen's *d* values (with positive values indicating boys have larger mean levels than girls and vice versa). Effect sizes are emphasized over statistical significance since our sample sizes are large and thus significance is expected for many relationships. Next, Pearson correlations between the subscales were computed for each age and cohort, separately by gender.

To formally test gender interactions, as well as to assess the effect size (betas) and amount of variance ($R^2$ values) in aggression explained by co-occurring behaviors, we ran linear regression models separately for each age and cohort. Before modeling, the subscale scores were standardized (mean = 0, SD = 1) to allow for comparability of the same questionnaire across different ages or cohorts where possible. Initial modeling included aggressive behavior as the dependent variable and one subscale score as the independent variable. There were three types of subscale scores in the initial models: 1) the externalizing problems subscale(s) with the highest correlation with aggressive behavior (e.g., hyperactivity); 2) the internalizing problems subscale(s) with the highest correlation with aggressive behavior (e.g., depression); and 3) the prosocial subscale (if available). For each model, data from boys and girls were modeled separately, after testing for gender interactions. Lastly, to examine multiple co-occuring behaviors with aggression,

regression modeling was performed with aggressive behavior as the dependent variable and the two (or three, if available) subscale measures from the initial models simultaneously modeled, modeled separately for gender after testing for gender interactions.

## Results

We examined 39,936 teacher ratings on children: 3627 observations from FT12, 4512 observations from GENR, 18,569 observations from NTR, and 13,228 observations from TEDS. There were 17,267 observations of 7-year-olds (49.3% boys), 2762 of 9-year-olds (46.9% boys), 6582 of 10-year-olds (49.6% boys), 11,884 of 12-year-olds (48.4% boys), and 1441 of 14-year-olds (48.4% boys). An interactive summary of all results can be found at (http://www.action-euproject.eu/TeacherRatingsChildAggression).

### Mean scores

Across all cohorts, mean levels were higher for boys than girls in all externalizing subscale scores, with Cohen's *d* values ranging 0.31–0.69 (Fig 1A–1C and S1 Table).

Results regarding internalizing problems were somewhat cohort dependent, with smaller gender effect sizes than for externalizing problems and prosocial behaviors (Fig 1A–1C and S1 Table). In FT12, girls had slightly higher levels of social anxiety compared to boys (Cohen's *d* ranged from -0.21 to -0.23), with boys and girls being more similar for depression (Cohen's *d* ranged from -0.09 to -0.12). In GENR, NTR and TEDS, for all internalizing problems, boys and girls generally had similar levels (Cohen's *d* ranged from -0.09 to 0.12).

For all cohorts, mean differences between boys and girls were found in all social behaviors, with girls having higher prosocial scores (Cohen's *d* ranged from -0.37 to -0.63), and boys having slightly more social/peer problems (Cohen's *d* range: 0.10–0.24).

### Co-occurrence of aggression and other behaviors and problems

In general, correlation patterns of aggressive behavior with other co-occurring behaviors were similar between genders and across cohorts and ages (Fig 2A–2C and S2 Table).

The strength of the correlations of aggressive behavior with other externalizing problems was substantial (Fig 2A–2C and S2 Table). Correlations between aggressive behavior and hyperactivity/attention problems ranged 0.51–0.75 for boys and 0.47–0.70 for girls. Correlations of aggressive behavior and rule-breaking behavior (from GENR and NTR) ranged 0.71–0.80 for boys and 0.63–0.74 for girls.

Correlations of aggressive behavior and internalizing problems ranged from small to moderate (Fig 2A–2C and S2 Table). Regarding depressive symptoms, correlations with aggressive behavior ranged 0.16–0.27 for boys and 0.15–0.29 for girls. Correlations of general anxiety problems with aggressive behavior (from GENR, NTR, and TEDS) ranged 0.17–0.39 in boys and 0.10–0.33 in girls, whereas social anxiety and aggressive behavior (from FT12) correlations ranged from -0.02 to -0.10 for boys and girls.

Correlations of aggressive behavior and social behaviors were moderate (Fig 2A–2C and S2 Table). Prosocial and aggressive behavior correlations (from FT12 and TEDS) ranged from -0.44 to -0.54 for boys and -0.33 to -0.44 for girls. Aggressive behavior and social/peer problems (from GENR, NTR, and TEDS) ranged 0.27–0.65 for boys and 0.26–0.65 for girls.

### Regression modeling

In linear regression modeling (i.e., aggressive behavior as the dependent and one subscale as the independent variable), gender was significantly associated with aggression. Furthermore,

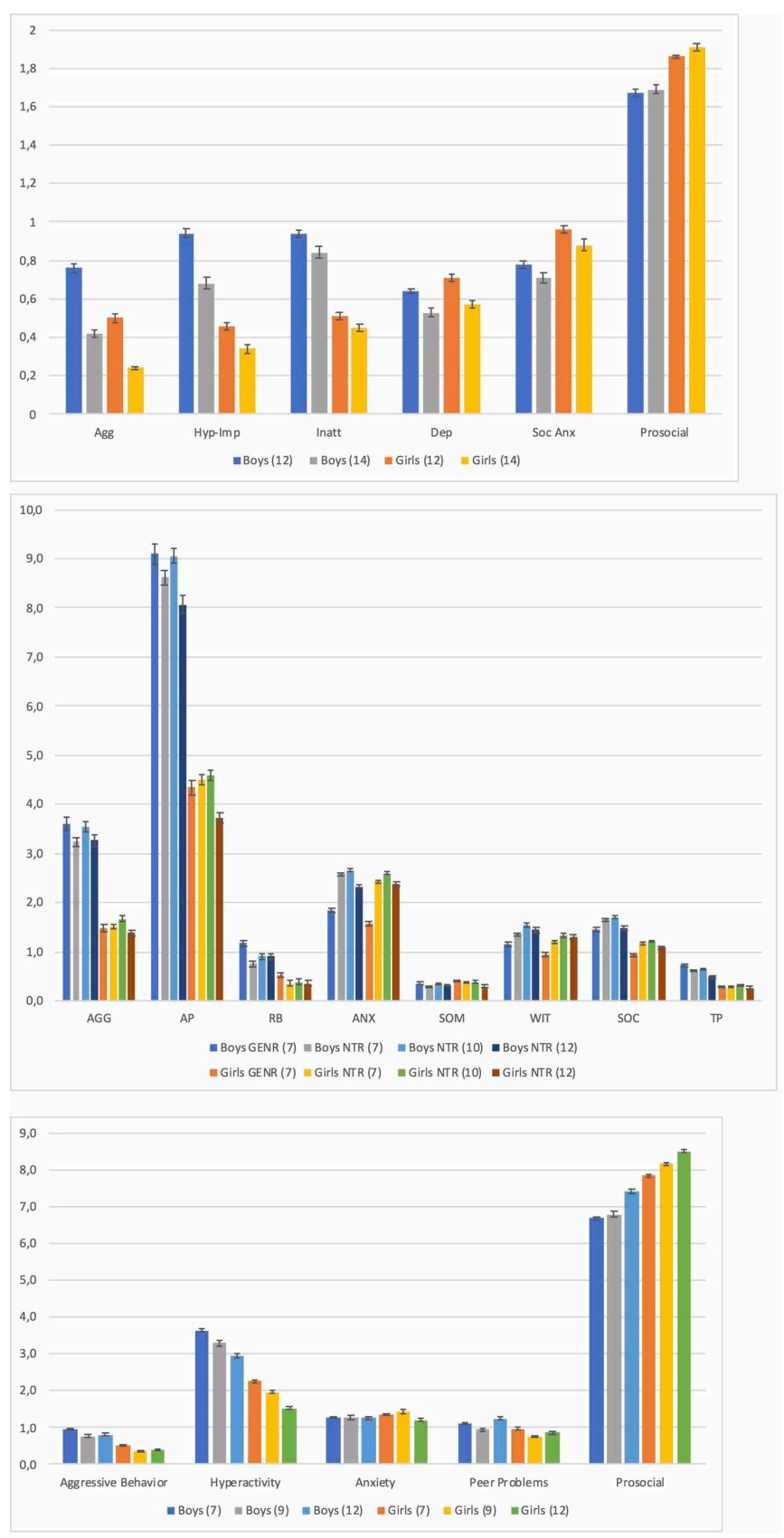

**Fig 1.** (a-c) Means and standard errors (SEs) of behavioral scales, separated by gender, age, cohort and behavioral questionnaire. a. MPNI questionnaire (FT12); Agg = aggression, Dep = depression, Hyp-Imp = hyperactivity-impulsivity, Inatt = inattention, Soc Anx = social anxiety. b. TRF questionnaire (GENR and NTR); AGG = aggression, ANX = anxious/depressed, AP = attention problems, RB = rule-breaking, SOC = social problems, SOM = somatic problems, TP = thought problems, WIT = withdrawn/depressed. c. SDQ questionnaire (TEDS).

all GENR and NTR and nearly all TEDS models indicated significant gender interaction terms (Table 1A–1C). In contrast, half of the FT12 models did not indicate significant gender interaction. Regarding model $R^2$ values (% of variance in dependent variable explained by independent variables), externalizing models were consistently largest (boy range: 0.26–0.63; girl range: 0.22–0.54). For internalizing models, $R^2$ values were small (boy range: 0.03–0.15; girl range: 0.01–0.11). For prosocial models, $R^2$ values ranged 0.11–0.29 (boy range: 0.19–0.29; girl range: 0.11–0.19).

In further regression modeling (i.e., multiple subscales included simultaneously as independent variables), gender interactions were generally significant. $R^2$ values generally did not differ from those in externalizing-only models (S3 Table). However, independent variables in these analyses were generally all significant, though with smaller effect sizes compared to initial models.

## Discussion

This study provides a summary of average levels and co-occurence of teacher-rated aggressive behavior and other behaviors and emotional problems for schoolchildren ages 7–14 in the European school setting. Although it is commonly assumed that aggressive behavior at school is associated with these other problems and behaviors, this is one of the few studies to look at large, population-based samples with teacher ratings of behaviors across multiple European countries. Results indicate that teacher-rated behavior patterns are quite similar across the different cohorts of Finland, the Netherlands, and the UK. We show that, as expected, the levels of aggressive behavior are statistically significantly different by gender, however, the effect sizes are only moderate. We also see that aggression often co-occurs with other behavioral and emotional difficulties, in both boys and girls, and that these correlations are quite similar between the genders. Regression modeling indicated that much of the variation ($R^2$) in aggressive behavior levels was explained by other co-occurring externalizing problems, though prosocial models also had rather large $R^2$ values in initial individual models. Modeling multiple co-occurring behaviors simultaneously indicated that children with aggressive behavior often have not only one co-occurring problem behavior, but multiple co-occurring problems (including low prosocial skills).

Regarding co-occurence of teacher-rated aggressive behavior with other externalizing problems, we see consistent similarities in the gender patterns. Although boys had, as expected, higher levels and associations of aggression and other externalizing problems compared to girls, correlations between aggression and other externalizing behaviors were very similar (*r* differences ≤0.1 between boys and girls). In this respect, we can point to another NTR cohort study regarding gender differences in ADHD diagnosis and comorbidity using teacher ratings [20] in which boys and girls had similar comorbidity profiles and school impairment, however, girls were less likely than boys to be identified by teachers as disruptive and referred for treatment. Similarly, conduct disorder and oppositional defiant disorder have both been more likely to be attributed to boys than girls among teacher ratings, compared to parent ratings [44, 45]. Previous studies have noted valid rater differences due to situational aggressive behavior [12], reflected in known low inter-rater agreement [9–11]. However, Stanger et al. [46] indicated that teacher ratings of externalizing problems alone were capable of predicting

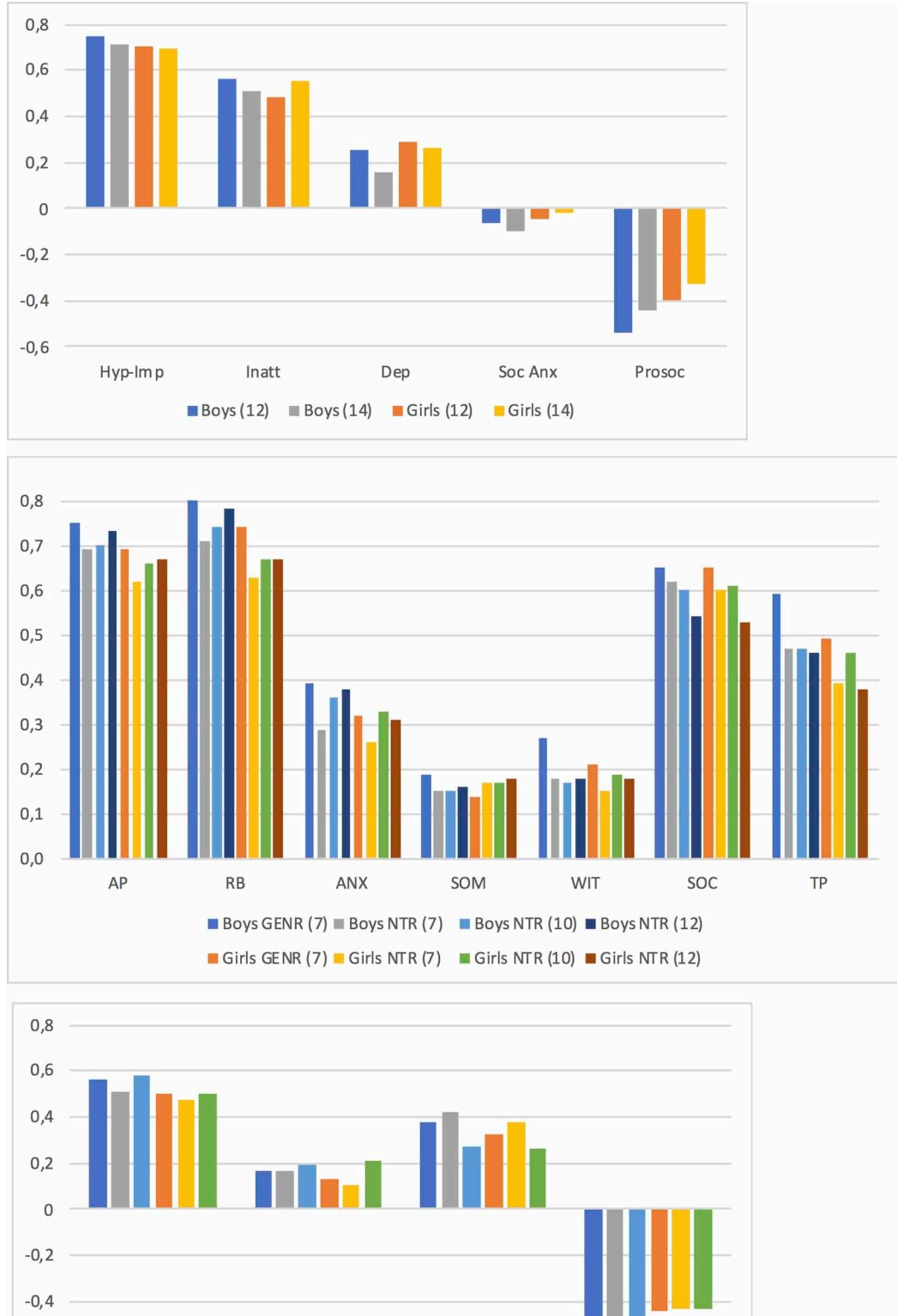

**Fig 2.** (a-c) Pearson correlations between aggression and other behavioral/emotional problems, separated by gender, age, cohort and behavioral questionnaire. a. MPNI questionnaire (FT12); Dep = depression, Hyp-Imp = hyperactivity-impulsivity, Inatt = inattention, Prosoc = prosocial, Soc Anx = social anxiety. b. TRF questionnaire (GENR and NTR); AGG = aggression, ANX = anxious/depressed, AP = attention problems, RB = rule-breaking, SOC = social problems, SOM = somatic problems, TP = thought problems, WIT = withdrawn/depressed. c. SDQ questionnaire (TEDS).

mental health referral (compared to combining with parental ratings). Regarding gender, however, Verhulst et al. [10] did note that for girls (and for externalizing problems), adding parental ratings to teacher ratings did improve poor outcome prediction over teacher ratings alone. Thus, for girls, multiple informant reports for externalizing behavior would be especially important, perhaps because of situational aggressive behavior difference. Additionally, increased awareness among teachers of girls' and boys' similar co-occuring behavior patterns in school could possibly improve girls' referral for treatment.

While the relationships between teacher-rated aggressive behavior and co-occurring externalizing problems are well-established, the co-occurrence with internalizing problems is less well described. Generally, there is weak to moderate correlation between teacher-rated aggressive behavior and depressive symptoms [21–23] as well as between aggressive behavior and general anxiety [23, 24], which we saw in our cohorts as well. For the association between aggressive behavior and social anxiety (in FT12), less well characterized in the literature, correlations were generally weak, non-significant, and negative. These mixed results regarding aggressive behavior and internalizing problems could partially be explained by August et al. [19] and Yang et al. [22] who found that, among schoolchildren, anxiety/mood disorders most often co-occur with multiple externalizing disorders (instead of, e.g., only conduct disorder). We also saw, in the multiple co-occuring behaviors models, that internalizing problems tend to co-exist alongside both aggressive behavior and other co-occuring externalizing behaviors. The ability of teachers to observe internalizing problems could play a role in the mixed results as well. Teachers are generally shown to be high in reliability regarding observing externalizing problems (e.g., intra-twin-pair correlations of monozygotic twins are 0.84) [13, 14], and while the reliability for internalizing is not as high (e.g., intra-twin-pair correlations of monozygotic twins are around 0.70), it remains satisfactory [13–15]. One note of observed similarity regarding aggressive behavior and internalizing problems was between boys and girls ($r$ differences ≤0.08 between boys and girls). Future studies should further clarify these results regarding aggressive behavior and internalizing problems, especially regarding anxiety, which can be a type of problem that is less often considered in the school setting than the more overt and administratively problematic externalizing behaviors.

Teacher-rated prosocial behavior was negatively associated with aggressive behavior, as has been observed by others [28], with moderate strength. In regression-based models, teacher-rated prosocial behavior explained much of the variance in aggressive behavior, but the effect of prosocial behavior attenuated substantially (though remained significant) when externalizing behavior was added to the model. The relationships are thus complex, aggressive students are often struggling not only with other externalizing problem behaviors but also a lack of social skills. It is of interest to note that Kokko and Pulkkinen [29] observed that teacher-rated prosocial behavior is protective in aggressive children against future unemployment. Additionally, Hämäläinen and Pulkkinen [47] have shown that peer-nominated aggressiveness without other co-occurring problems did not predict future criminality in men, while an accumulation of behavioral problems (including aggressive behavior and poor prosocial skills) predicted their future criminality best. Interestingly, they also showed that low peer-nominated prosocial behavior alone predicted future criminality. This could be related to issues of empathy (which has been positively linked to prosocial behavior and negatively linked to aggressive

**Table 1.** (a-c) Linear regression summaries of standardized single independent variable models. Modeled separately by behavioral questionnaire, age wave of data collection, and gender.

**a. MPNI questionnaire (FT12)**

| *Age 12 Boys (N = 1105)* | | | | *Age 12 Girls (N = 1081)* | | | | Gender interaction model | |
|---|---|---|---|---|---|---|---|---|---|
| **Model** | **Agg β** | **95% CI** | **R²** | **Model** | **Agg β** | **95% CI** | **R²** | **Interaction term, p** | **R²** |
| Hyp-Imp | 0.73 | 0.69, 0.77 | 0.56 | Hyp-Imp | 0.80 | 0.75, 0.85 | 0.48 | 0.03 | 0.55 |
| Depression | 0.27 | 0.21, 0.34 | 0.06 | Depression | 0.25 | 0.20, 0.30 | 0.08 | 0.53 | 0.11 |
| Prosocial | -0.57 | -0.62, -0.51 | 0.29 | Prosocial | -0.38 | -0.43, -0.32 | 0.16 | 0.00 | 0.27 |
| *Age 14 Boys (N = 697)* | | | | *Age 14 Girls (N = 744)* | | | | Gender interaction model | |
| **Model** | **Agg β** | **95% CI** | **R²** | **Model** | **Agg β** | **95% CI** | **R²** | **Interaction term, p** | **R²** |
| Hyp-Imp | 0.73 | 0.67, 0.78 | 0.50 | Hyp-Imp | 0.70 | 0.64, 0.75 | 0.48 | 0.46 | 0.51 |
| Depression | 0.20 | 0.11, 0.28 | 0.03 | Depression | 0.21 | 0.16, 0.27 | 0.07 | 0.73 | 0.07 |
| Prosocial | -0.50 | -0.57, -0.42 | 0.19 | Prosocial | -0.29 | -0.35, -0.23 | 0.11 | 0.00 | 0.19 |

**b. TRF questionnaire (GENR and NTR)[a]**

| **GEN-R** *Age 7 Boys (N = 2270)* | | | | *Age 7 Girls (N = 2242)* | | | | Gender interaction model | |
|---|---|---|---|---|---|---|---|---|---|
| **Model** | **Agg β** | **95% CI** | **R²** | **Model** | **Agg β** | **95% CI** | **R²** | **Interaction term, p** | **R²** |
| Attention Problems | 0.80 | 0.77, 0.83 | 0.56 | Attention Problems | 0.62 | 0.59, 0.65 | 0.48 | <0.001 | 0.56 |
| Rule-Breaking | 0.81 | 0.78, 0.83 | 0.63 | Rule-Breaking | 0.68 | 0.66, 0.71 | 0.54 | <0.001 | 0.63 |
| Anxious/Depressed | 0.45 | 0.40, 0.49 | 0.15 | Anxious/Depressed | 0.22 | 0.19, 0.25 | 0.10 | <0.001 | 0.17 |
| **NTR** *Age 7 Boys (N = 3416)* | | | | *Age 7 Girls (N = 3518)* | | | | Gender interaction model | |
| **Model** | **Agg β** | **95% CI** | **R²** | **Model** | **Agg β** | **95% CI** | **R²** | **Interaction term, p** | **R²** |
| Attention Problems | 0.74 | 0.71, 0.76 | 0.48 | Attention Problems | 0.56 | 0.54, 0.58 | 0.38 | <0.001 | 0.48 |
| Rule-Breaking | 0.72 | 0.70, 0.75 | 0.51 | Rule-Breaking | 0.57 | 0.55, 0.60 | 0.40 | <0.001 | 0.50 |
| Anxious/Depressed | 0.34 | 0.30, 0.37 | 0.09 | Anxious/Depressed | 0.19 | 0.17, 0.21 | 0.07 | <0.001 | 0.12 |
| *Age 10 Boys (N = 3264)* | | | | *Age 10 Girls (N = 3318)* | | | | Gender interaction model | |
| **Model** | **Agg β** | **95% CI** | **R²** | **Model** | **Agg β** | **95% CI** | **R²** | **Interaction term, p** | **R²** |
| Attention Problems | 0.73 | 0.71, 0.76 | 0.48 | Attention Problems | 0.62 | 0.59, 0.64 | 0.43 | <0.001 | 0.49 |
| Rule-Breaking | 0.73 | 0.71, 0.75 | 0.55 | Rule-Breaking | 0.67 | 0.65, 0.70 | 0.45 | 0.004 | 0.54 |
| Anxious/Depressed | 0.43 | 0.39, 0.46 | 0.13 | Anxious/Depressed | 0.24 | 0.22, 0.26 | 0.11 | <0.001 | 0.16 |
| *Age 12 Boys (N = 2477)* | | | | *Age 12 Girls (N = 2576)* | | | | Gender interaction model | |
| **Model** | **Agg β** | **95% CI** | **R²** | **Model** | **Agg β** | **95% CI** | **R²** | **Interaction term, p** | **R²** |
| Attention Problems | 0.77 | 0.75, 0.80 | 0.54 | Attention Problems | 0.62 | 0.59, 0.65 | 0.45 | <0.001 | 0.54 |
| Rule-Breaking | 0.77 | 0.75, 0.80 | 0.61 | Rule-Breaking | 0.67 | 0.64, 0.70 | 0.45 | <0.001 | 0.59 |
| Anxious/Depressed | 0.47 | 0.42, 0.51 | 0.14 | Anxious/Depressed | 0.2 | 0.18, 0.23 | 0.10 | <0.001 | 0.17 |

**c. SDQ questionnaire (TEDS)**

| *Age 7 Boys (N = 2834)* | | | | *Age 7 Girls (N = 2987)* | | | | Gender interaction model | |
|---|---|---|---|---|---|---|---|---|---|
| **Model** | **Agg β** | **95% CI** | **R²** | **Model** | **Agg β** | **95% CI** | **R²** | **Interaction term, p** | **R²** |
| Hyperactivity | 0.59 | 0.56, 0.63 | 0.31 | Hyperactivity | 0.49 | 0.46, 0.52 | 0.25 | <0.001 | 0.31 |
| Anxiety | 0.20 | 0.16, 0.25 | 0.03 | Anxiety | 0.11 | 0.08, 0.14 | 0.02 | <0.001 | 0.05 |
| Prosocial | -0.56 | -0.59, -0.52 | 0.25 | Prosocial | -0.40 | -0.43, -0.37 | 0.19 | <0.001 | 0.25 |
| *Age 9 Boys (N = 1295)* | | | | *Age 9 Girls (N = 1467)* | | | | Gender interaction model | |
| **Model** | **Agg β** | **95% CI** | **R²** | **Model** | **Agg β** | **95% CI** | **R²** | **Interaction term, p** | **R²** |
| Hyperactivity | 0.57 | 0.52, 0.62 | 0.26 | Hyperactivity | 0.42 | 0.37, 0.46 | 0.22 | <0.001 | 0.27 |
| Anxiety | 0.21 | 0.14, 0.27 | 0.03 | Anxiety | 0.07 | 0.04, 0.11 | 0.01 | <0.001 | 0.05 |
| Prosocial | -0.56 | -0.62, -0.51 | 0.25 | Prosocial | -0.36 | -0.40, -0.32 | 0.18 | <0.001 | 0.25 |
| *Age 12 Boys (N = 2168)* | | | | *Age 12 Girls (N = 2477)* | | | | Gender interaction model | |
| **Model** | **Agg β** | **95% CI** | **R²** | **Model** | **Agg β** | **95% CI** | **R²** | **Interaction term, p** | **R²** |
| Hyperactivity | 0.59 | 0.56, 0.63 | 0.34 | Hyperactivity | 0.50 | 0.46, 0.53 | 0.25 | <0.001 | 0.33 |
| Anxiety | 0.21 | 0.17, 0.26 | 0.04 | Anxiety | 0.17 | 0.14, 0.20 | 0.05 | 0.092 | 0.07 |

(*Continued*)

**Table 1.** (Continued)

| Prosocial | -0.52 | -0.56, -0.48 | 0.23 | Prosocial | -0.38 | -0.41, -0.35 | 0.19 | <0.001 | 0.24 |
|---|---|---|---|---|---|---|---|---|---|

[a]Both 'Attention Problems' and 'Rule-Breaking' are modeled due to 'Attention Problems' being more comparable to the other cohorts' modeled externalizing behavior measure, although 'Rule-Breaking' was the externalizing behavior most highly correlated with aggression (a specified 'rule' in the methods for inclusion in linear regression).

Abbreviations: Agg = aggression, CI = confidence interval, Hyp-Imp = hyperactivity–impulsivity.

behavior [48]) or suggestions to subtype prosocial behavior (e.g., instrumental prosocial behavior) [30], with studies having shown differences in public/instrumental prosocial behavior being associated with aggression and lack of empathy while more anonymous/non-instrumental prosocial behavior is negatively associated [30, 49]. Regarding gender differences, we found that prosocial–aggressive behavior correlations were less than 0.15 different between boys and girls, and the differences between genders in TEDS correlation values were half of those in FT12. This may be related to the aggression measure, since the SDQ (used in TEDS) captures conduct problems, which are more heterogenous (combines aspects of aggressive and rule-breaking behaviors) than the general aggression captured by the MPNI (used in FT12), thus suggesting that, especially when comparing anti-social aspects of aggression with levels of prosocial behavior, there are no gender differences.

In our study protocol, each age from each cohort represents a cross-sectional snapshot of behavioral and emotional problems, with some children captured in more than one age category. Because all data are drawn in large numbers and in a population-based manner, these data do reflect developmental stages well. Over the ages (7–14 years), we can see that patterns do not generally differ by gender. For all cohorts, the differences in aggressive–co-occurring behavior correlations between all ages were <0.11. In longitudinal settings, the general trend is that population levels of aggressive behavior diminish as age increases, although there tends to be stability in rank order across certain individuals (those with high aggression and those of lower socioeconomic status) [8, 50].

Although detailed comparison between cohorts regarding behavioral questionnaires, countries and school systems are outside the scope of this paper (however, see [51] and S1 Text, for information on these aspects in the cohorts and countries), it is noteworthy that we see similar patterns across cohorts (questionnaires), since the cohorts represent different European countries and the questionnaires were developed for different purposes. Indeed, future studies should consider validating their findings in other countries due to the robustness of the results across countries and questionnaires. Furthermore, both the ASEBA system (of which the TRF is a part) and the SDQ were assessed by Achenbach et al. [52] and generally found to produce comparable results across countries, with more variation found within populations than between. Moreover, in ACTION consortium analyses, the SDQ and CBCL (parent version of the ASEBA questionnaire family of which the TRF is a part) aggression scales were found to capture the same underlying genetic aspects of aggression [53], despite item-level differences, and the negative relationship between aggression and academic performance was consistent and generally similar in FT12, NTR and TEDS cohorts using teacher-rated MPNI, SDQ, and TRF [54].

One final discussion considers a parallel analysis among the same ACTION cohorts, using parental and self ratings ([5]; interactive results also available at http://www.action-euproject. eu/ComorbidityChildAggression). Only rough comparions can be made between the studies although the protocols are similar for both studies, the sample sizes for cohorts are large, and the children are generally the same in both studies. Overall, teacher-based correlations of

aggressive behavior with co-occuring behaviors are higher for externalizing problems than parental ratings at similar ages. Conversely, our teacher-based co-occurrence correlations were lower for internalizing problems and prosocial behaviors than parental ratings at similar ages. In comparison, McConaughy et al. [24] also found teacher-rated co-occurrence of aggressive behavior and internalizing problems to be lower than parent-rated behaviors, however, they noted minimal difference between raters for co-occurrence correlations between aggressive behavior and other externalizing behaviors.

Despite the nearly 40,000 teacher-rated observations of behavioral and emotional problems among school children ages 7–14 collected across 3 European countries, there are limitations to consider in this study. The data come from three higher income countries in Europe, and it is unclear if these patterns would remain in lower income countries or countries of differing cultural backgrounds. For example, a recent publication on data from 63 low- and middle-income countries collected by the World Health Organization's Global School-based Student Health Survey indicates that gender differences in physical aggression are stronger as a function of greater gender equality in a society [33].

Another important limitation could be regarding teacher ratings themselves. For example, teachers may not be able to observe all behaviors in the school setting; however, ratings from teachers and peers (who can observe their classmates outside the range of teachers' observation) are among the most highly correlated of inter-rater pairs [9, 13], especially in early/middle adolescence (compared to late adolescence) [55]. Additionally, this study captures broadly defined aggressive behavior, which means it would be important to consider teachers' ability to observe a broad range of aggressive behavior. In Pakaslahti et al. [55], consistency was found to be higher across teacher, peer and self ratings of direct aggression (compared to indirect aggression), but no gender differences were found in teacher–peer rating comparisons of indirect aggression, suggesting that teachers can observe a wide range of aggressive behavior. Considering potential gender bias further, we can consider teachers compared to parents. Rescorla et al. [43] investigated the consistency of the TRF across 21 countries and discussed findings in light of a previous related study on the CBCL. For externalizing behaviors, boys had consistently higher scores than girls in the TRF (same in the CBCL), while no significant within-country gender effect was found for internalizing problems for ages 6–11 in the TRF (the main age range of our study), compared to girls often having significantly higher scores than boys on the CBCL. Rescorla et al. [43] suggest that teachers may be equally unaware of internalizing problems of boys and girls (leading to limited apparent gender bias; see also [56]), whereas daughters may be more likely to share their emotions with their parents than boys (leading to a potential gender bias in parent ratings). In summary, although teachers may not be able to capture all aspects of child behaviors and emotional problems at school in an unbiased manner, they are a valuable resource and a relatively convenient informant from whom to collect data.

Lastly, three of the four cohorts consist of twins. While twins are born on average prematurely and of lower birthweight, they are generally indistinguishable from singletons later in childhood on multiple traits and conditions. Studies from, for example, FT12 and NTR [57, 58] have shown that the twins are representative of schoolchildren with respect to both internalizing and externalizing behaviors and educational achievement. Additionally, twins are born in every stratum of society, and twins' parents and teachers are generally well motivated to take part in research.

We have presented a summary of teacher-rated childhood aggressive behavior levels and associations with co-occurring behaviors and emotional problems in the school setting. These results indicate that aggressive behavior in school regularly co-occurs with other externalizing behaviors and a lack of prosocial skills, and moderately co-occurs with internalizing problems.

Furthermore, we draw attention to the relative similarities in patterns of associations between aggressive behavior and co-occurring behaviors across genders and participating cohorts using different instruments. Teachers are a valuable resource for characterizing children's behaviors, especially externalizing and prosocial behaviors, and we see that although there may be gender differences in separate behavior scales, teacher ratings do not indicate strong gender differences in co-occuring behaviors. However, teacher trainings could help to reduce potential gender bias regarding problem behaviors and to recognize that those with one problem behavior possibly have multiple problem behaviors they are struggling with. Additionally, school interventions for aggression need to be holistic, focusing on broad behavioral and emotional improvement including support to develop prosocial skills, such as the successful "multi-year universal social–emotional learning program" implemented by Greenberg et al. [59] that showed reduced levels of aggression and increased prosocial skills.

## Supporting information

**S1 Text. Supplemental text regarding detailed teacher rating collections and the school systems in Finland, the Netherlands, and the UK.**
(DOCX)

**S1 Table.** Mean, standard deviation (SD), standard error (SE) and effect sizea summaries by behavioral questionnaire, age, and gender; A. MPNI questionnaire (FT12)b, B. TRF questionnaire (GENR and NTR)b, C. SDQ questionnaire (TEDS)b.
(DOCX)

**S2 Table.** Pearson correlations of all behavioral subscales with each other by behavioral questionnaire, age, and gender; A. MPNI questionnaire (FT12)*, B. TRF questionnaire (GENR and NTR)*; C. SDQ questionnaire (TEDS).
(DOCX)

**S3 Table. Linear regression summaries of standardized multiple independent variable models.** Modeled separately by behavioral questionnaire, age wave of data collection, and gender; A. MPNI questionnaire (FT12), B. TRF questionnaire (GENR and NTR), C. SDQ questionnaire (TEDS).
(DOCX)

## Acknowledgments

FT12: We gratefully acknowledge the ongoing contribution of the participating twin families.

GENR: The authors gratefully acknowledge the contribution of all children and parents, general practitioners, hospitals, midwives, and pharmacies involved in the Generation R Study. The Generation R Study is conducted by the Erasmus Medical Center (Rotterdam) in close collaboration with the School of Law and Faculty of Social Sciences of the Erasmus University Rotterdam; the Municipal Health Service Rotterdam area, Rotterdam; the Rotterdam Homecare Foundation, Rotterdam; and the Stichting Trombosedienst & Artsenlaboratorium Rijnmond, Rotterdam.

NTR: We gratefully acknowledge the ongoing contribution of the participating families.

TEDS: We gratefully acknowledge the ongoing contribution of the participants in the Twins Early Development Study (TEDS) and their families.

## Author Contributions

**Conceptualization:** Meike Bartels, Dorret I. Boomsma.

**Data curation:** Alyce M. Whipp, Koen Bolhuis, Eveline L. de Zeeuw, Richard J. Rose, Catharina (Toos) E. M. van Beijsterveldt, Robert Plomin, Henning Tiemeier.

**Formal analysis:** Alyce M. Whipp, Koen Bolhuis.

**Investigation:** Alyce M. Whipp, Eero Vuoksimaa, Dorret I. Boomsma.

**Methodology:** Alyce M. Whipp, Eero Vuoksimaa, Tellervo Korhonen, Lea Pulkkinen, Kaili Rimfeld, Meike Bartels, Jaakko Kaprio, Dorret I. Boomsma.

**Resources:** Eveline L. de Zeeuw, Matteo Mauri.

**Supervision:** Eero Vuoksimaa, Tellervo Korhonen, Lea Pulkkinen, Jaakko Kaprio, Dorret I. Boomsma.

**Visualization:** Matteo Mauri.

**Writing – original draft:** Alyce M. Whipp, Eero Vuoksimaa.

**Writing – review & editing:** Alyce M. Whipp, Eero Vuoksimaa, Koen Bolhuis, Eveline L. de Zeeuw, Tellervo Korhonen, Matteo Mauri, Lea Pulkkinen, Kaili Rimfeld, Richard J. Rose, Catharina (Toos) E. M. van Beijsterveldt, Meike Bartels, Robert Plomin, Henning Tiemeier, Jaakko Kaprio, Dorret I. Boomsma.

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
