## [Decision Letter · Decision Letter 0]

4 Dec 2020

PONE-D-20-29433

Teacher-rated aggression and co-occurring behaviors and problems among schoolchildren: A comparison of four population-based European cohorts

PLOS ONE

Dear Dr. Whipp

Thank you for submitting your manuscript to PLOS ONE. After careful consideration, we feel that it has merit but does not fully meet PLOS ONE’s publication criteria as it currently stands. Therefore, we invite you to submit a revised version of the manuscript that addresses the points raised during the review process. Kindly amend as recommended by the two reviewers and indicate where you have revised the text to confirm this in your re-submission.

Please submit your revised manuscript by January 4, 2021. If you will need more time than this to complete your revisions, please reply to this message or contact the journal office at plosone@plos.org. Please include the following items when submitting your revised manuscript:

We look forward to receiving your revised manuscript.

Kind regards,

Gerard Hutchinson, MD

Academic Editor

PLOS ONE

Journal Requirements:

Reviewers' comments:

Reviewer's Responses to Questions

**Comments to the Author**

1. Is the manuscript technically sound, and do the data support the conclusions?

Reviewer #1: Partly

Reviewer #2: Partly

2. Has the statistical analysis been performed appropriately and rigorously? 

Reviewer #1: Yes

Reviewer #2: Yes

3. Have the authors made all data underlying the findings in their manuscript fully available?

Reviewer #1: No

Reviewer #2: Yes

4. Is the manuscript presented in an intelligible fashion and written in standard English?

Reviewer #1: No

Reviewer #2: Yes

5. Review Comments to the Author

Reviewer #1: Thank you for the opportunity to revise the manuscript “Teacher-rated aggression and co-occurring behaviors and problems among schoolchildren: A comparison of four population-based European cohorts”. The aim of the study was to investigate the co-occurring behaviors which are related to aggression in the school setting, using teacher ratings of children ages 7–14, while assessing gender differences in their mean level.

Please find below my comments.

1) Have the authors made all data underlying the findings in their manuscript fully available?

It is stated that the data that support the findings of the study are not publicly available due to privacy or ethical restrictions. The corresponding author is able to provide analysis code and information on how to access the cohorts and request the data from the appropriate managers of each cohort.

2) Is the manuscript presented in an intelligible fashion and written in standard English?

I found some typographical and grammatical errors

3) The authors provided reference for a previous work in which, based on my understanding, the same data were used for testing remarkably similar research questions (i.e., http://www.action-euproject.eu/ComorbidityChildAggression). The main element of distinction from this previous work should be, in principle, the use of the teachers’ rating (rather than parental and self-reported behaviors) in assessing different students’ behavioral characteristics (i.e., aggression, externalizing, internalizing, and prosocial behavior). However, I think the justification for this additional work should be strengthen in the manuscript. First, what is the additional value of using teacher ratings? In the introduction, page 8, the authors stated, “Teachers’ ability to observe children in a complex task-based setting and among peers of similar ages and abilities provides them with a valuable comparison-base, making their insight unique and important”. Is, therefore, the teacher rating selected because of its supposed accuracy in identifying different behavioral characteristics of the students in the school setting? But school aggression can often take place out of the teacher sight. Additionally, in the paper teacher ratings were used also for assessing the co-occurrent behaviors of interest. Is it safe to assume that teacher can be equally able to assess externalizing behaviors (perhaps more visible) and internalizing behaviors, which perhaps might require more sensitivity, attention, and diagnostic skills? Most importantly, what about the potential biases in teachers’ perceptions? From my perspective, what the paper is actually measuring, and testing are the characteristics that teachers tend to associate with aggression, rather than actual behaviors. Teachers might formulate their evaluations about aggression based on other criteria than the actual behavior of the students. These criteria might be derived from cognitive assumptions (see the halo effect, i.e., teacher might attribute several negative characteristics, and less positive characteristics, to students who are perceived as aggressive) as well as previous experience of what is more likely (a statistical association between aggression and, let’s say, gender). I think these aspects should be addressed and much more problematize over the manuscript (also in the way in which the authors frame their conclusions), in order to best assess the rationale of the study and the substantial additional value of focusing on the perspective of the teachers, especially in comparation to the previous existing work.

4) I think the way in which the title is current specified to be quite confusing, as it is mentioned a comparation between 4 different cohorts (and 3 different national contexts). However, this comparative element was not addressed in a structural and empirically proper way. And the author themselves, page 20 in the conclusion section stated, “Although detailed comparison between cohorts regarding behavioral questionnaires, countries and school systems are outside the scope of this paper”. If is that the case, and a structural comparation is challenging given, in the first place, the differences between scales in different cohorts, I would suggest removing the term “comparison” from the title and caution when formulating considerations about similarity in patterns, as it is not in line with what has been statistically tested.

5) It is not totally clear to me on which type of aggression the authors are focusing on. In the introduction they provide the distinction between 4 subcategories, namely: proactive, reactive, direct, and indirect. Is the focus of the paper equally on all of them, or just some? Are all these subdimensions caught by the different scales used in the operazionalization? Is there homogeneity in the 4 questionnaires (i.e., are they referring to the same construct)? Concerning this, I think it would be helpful for the readers to provide more directly in the text the different items which were used for computing the aggression measure in each of the 4 questionnaires.

6) I would provide in the introduction a more clear and complete discussion about existing literature and findings especially concerning the association between aggression and the different co-occurrent behaviors under investigation. I missed especially what are the existing findings, if any, concerning the relation between aggression and prosocial behavior.

7) I am curious why the author refers to the regression models as “supplemental’ material, as it is supposed to be a central part of the analyses. I would modify this in the text, and probably suggest including the tables in the manuscript rather than as online resource, as this can be the most informative output. Also, I am wondering why the author decided to present both the interaction and the differentiated analyses by gender. I personally find this quite redundant, as the interactions alone should be enough informative.

Reviewer #2: Thank you very much for this interesting paper. The authors show results on the epidemiology of teacher rated aggressive behavior in school children based on an impressive data set. This paper uses a large, international sample which gives interesting findings on

mean levels of teacher rated aggressive behavior and its associations with internalizing / externalizing behavior and prosocial behavior in 7- to 14-year-old students. It seems that this valuable data has also been analyzed in a longitudinal manner in other papers.

However, bevor publishing I highly suggest to discuss the following major aspects more deeply:

Abstract:

I suggest to explicitly mention the overall aim of this study within the abstract.

I highly recommend to define aggressive behavior within the abstract. As you mention within your introduction there are different types of aggressive behavior. However, the questionnaires you used may assess only parts of the aggression construct.

Please include information on sample age and school types (if applicable).

Introduction:

As mentioned above, you correctly describe the more complex construct of aggression. However, I do not see that you actually assessed all subtypes. Please specify what kind of aggression you are referring to. In addition, some subtypes may be assessed less valid and reliable within a teacher rating (e.g. relational/ indirect aggression). I suggest to include literature on what kind of aggressive behavior can be reliably assessed though teacher ratings. Also, consider to mention this in your limitation section.

Page 8, second sentence “Furthermore, children with aggression and comorbidities often have more than one comorbidity and worse prognosis and poorer outcomes …”. Please specify, worse than what? Poorer outcomes with respect to what?

Page 8, third sentence (“…regarding the school setting”). Please underlie your statement with literature. What about literature on bullying within schools. In my understanding most studies rely on population-based samples. There are few studies comparing population-based samples with clinical samples (see FemNAT-CD study as one recent example including clinical samples).

Materials and methods

Please specify „handicap or illness that interferes with daily functioning“. How has this been assessed?

Did the excluded sample due to missing values differ to the included sample with respect to any characteristics? E.g., IQ, SES, teacher or child characteristics, internalizing/externalizing behavior?

Cohort description:

Do you have any background information on the samples: IQ, SES, single parent household, psychiatric history?

Do the samples differ with respect to IQ, SES, etc.

If I understood correctly you included some children twice in your study for different age groups, as your data stems from longitudinal studies. Do results differ, if you include each child only once? Can you control for this in your model?

It would be highly interesting to use teacher ratings on aggression as a predictor to later aggressive behavior longitudinally.

How did you get teacher information? For how long did the teacher know the child? Do you have any information on the teachers and schools?

Study questionnaires

Please consider sharing information on the questionnaires and subscales psychometrics (Conbachs alpha etc.,)

I suggest to give more information on what kind of aggressive behavior has been assessed through those questionnaires. Please give item examples.

In my understanding conduct problems include more behavioristics than just aggression. Please specify.

Analysis

Please consider to control for study site. Especially, as you mention later, that gender differences in internalizing subscale differed between studies. Also, I suggest to control for IQ.

Did you include aggression items within the overall externalizing problem scale? If so, do I understand correctly, that you include items on aggression both in your dependent and independent variables?

How exactly did you test for gender interaction? What was your model?

Results

Please check with the journal specifications: I suggest to include r = and N =, etc.

Page 15, information on gender differences in internalizing problems compared between studies: I suggest to discuss the different findings. This may be a reason to use site as random effect within the model.

Discussion

I suggest to discuss more deeply the following aspects:

- What types of aggression are the results referring to? I suggest to look into the following literature: Ackermann, K., Kirchner, M., Bernhard, A. et al. Relational Aggression in Adolescents with Conduct Disorder: Sex Differences and Behavioral Correlates. J Abnorm Child Psychol 47, 1625–1637 (2019). https://doi.org/10.1007/s10802-019-00541-6

- Do you think teacher reports on internalizing behavior are reliable? May it be more appropriate to use self-ratings for internalizing behavior? Do you have any self or parent ratings with which you could compare results?

- I agree that teachers are a valuable and most likely the most reliable informant on aggressive behavior. However, can you state the following based on your results? (“However, girls were less likely than boys to be identified by teachers as disruptive and referred for treatment “.) Can you say that, although you did not compare with self- and parent ratings? Maybe girls are just less aggressive? Maybe, because relational aggression is less likely to be observed by teachers/ others?

- In addition, page 22 (“Teachers are a valuable resource for identifying children in need of specific support and possibly in referral or diagnostic aspects”). I agree, but can you say this along this data? How can you make sure that they capture most children in need of psychological help as you did not compare with a clinical sample?

- Could the teacher rating explain the moderate correlation between aggression and internalizing behavior? As internalizing behavior may not be as visible (cognitions rather than behavior) as externalizing behavior?

- You describe aggression dimensional and not on a clinical level. Therefore, are you able to show with your results that higher scores on internalizing behavior may be a „protective factor“ for externalizing behavior? Is there literature to this?

- You mention highly interesting results on prosocial behavior and future outcomes. Consider to discuss literature on CU traits to this highly interesting finding.

- Page 21. Please explain the following statement: “Overall, teacher-based correlations of aggressive behavior with co-occurring behaviors are higher for externalizing problems and lower for internalizing problems and prosocial behaviors than parental ratings at similar ages.“

6. PLOS authors have the option to publish the peer review history of their article (what does this mean?). If published, this will include your full peer review and any attached files.

Reviewer #1: No

Reviewer #2: No

---

## [Author Response · Author response to Decision Letter 0]

26 Feb 2021

all responses to the editor and peer reviewers are now in the 'Response to Reviewers' document in the submission

Journal Requirements:

Authors’ response: We have reviewed the above pdf links and PLOS ONE’s online submission guidelines. We hope we have now complied with the journal requirements. If there is more modification we need to do based on formatting/style, we will do them. 

Authors’ response: None of the cohorts are able to publicly share a de-identified dataset because the data contain potentially sensitive information and the participant consents and ethical committees have not approved that. The Generation R datasets analyzed during the current study are not publicly available due to the terms and conditions participants agree to when they participate in Generation R, but are available from data management on reasonable request (datamanagementgenr@erasmusmc.nl). The Twins Early Development Study’s data access policy can be found at the following link https://www.teds.ac.uk/researchers/teds-data-access-policy. Applications for data and information on the access policy for the Netherlands Twin Register may be found at: https://tweelingenregister.vu.nl/submitting-a-data-sharing-request. FT12 data are available through the Institute for Molecular Medicine Finland (FIMM) Data Access Committee (DAC) for authorized researchers who have IRB/ethics approval and an institutionally approved study plan. For more details, please contact the FIMM DAC (fimm-dac@helsinki.fi).

Authors’ response: This change has now been made. The ethics statement now only appears in the Methods section. 

Authors’ response: We have visited the ‘Supporting Information guidelines’ link and believe we have modified our files and in-text citations to comply, as well as included captions at the end of the manuscript.

Reviewer #1: Thank you for the opportunity to revise the manuscript “Teacher-rated aggression and co-occurring behaviors and problems among schoolchildren: A comparison of four population-based European cohorts”. The aim of the study was to investigate the co-occurring behaviors which are related to aggression in the school setting, using teacher ratings of children ages 7–14, while assessing gender differences in their mean level.

Please find below my comments.

1) Have the authors made all data underlying the findings in their manuscript fully available?

It is stated that the data that support the findings of the study are not publicly available due to privacy or ethical restrictions. The corresponding author is able to provide analysis code and information on how to access the cohorts and request the data from the appropriate managers of each cohort.

Authors’ response: We have provided more information regarding this in the above Journal Requirements #2 item above. 

2) Is the manuscript presented in an intelligible fashion and written in standard English?

I found some typographical and grammatical errors

Authors’ response: The final manuscript draft has been read through by the co-authors (with careful attention by the corresponding author) and we believe any small grammar or typographical errors have now been removed.

3) The authors provided reference for a previous work in which, based on my understanding, the same data were used for testing remarkably similar research questions (i.e., http://www.action-euproject.eu/ComorbidityChildAggression). The main element of distinction from this previous work should be, in principle, the use of the teachers’ rating (rather than parental and self-reported behaviors) in assessing different students’ behavioral characteristics (i.e., aggression, externalizing, internalizing, and prosocial behavior). However, I think the justification for this additional work should be strengthen in the manuscript. First, what is the additional value of using teacher ratings? In the introduction, page 8, the authors stated, “Teachers’ ability to observe children in a complex task-based setting and among peers of similar ages and abilities provides them with a valuable comparison-base, making their insight unique and important”. Is, therefore, the teacher rating selected because of its supposed accuracy in identifying different behavioral characteristics of the students in the school setting? But school aggression can often take place out of the teacher sight. Additionally, in the paper teacher ratings were used also for assessing the co-occurrent behaviors of interest. Is it safe to assume that teacher can be equally able to assess externalizing behaviors (perhaps more visible) and internalizing behaviors, which perhaps might require more sensitivity, attention, and diagnostic skills? Most importantly, what about the potential biases in teachers’ perceptions? From my perspective, what the paper is actually measuring, and testing are the characteristics that teachers tend to associate with aggression, rather than actual behaviors. Teachers might formulate their evaluations about aggression based on other criteria than the actual behavior of the students. These criteria might be derived from cognitive assumptions (see the halo effect, i.e., teacher might attribute several negative characteristics, and less positive characteristics, to students who are perceived as aggressive) as well as previous experience of what is more likely (a statistical association between aggression and, let’s say, gender). I think these aspects should be addressed and much more problematize over the manuscript (also in the way in which the authors frame their conclusions), in order to best assess the rationale of the study and the substantial additional value of focusing on the perspective of the teachers, especially in comparation to the previous existing work.

Authors’ response: Thank you for encouraging us to think about the overarching motivations for the study and the use of teacher ratings. We put our efforts into answering your questions in a satisfactory way, while also addressing related concerns by Reviewer 2 in their questions: “As mentioned above, you correctly describe the more complex construct of aggression…”, “Do you think teacher reports on internalizing behavior are reliable?...”, “Could the teacher rating explain the moderate correlation between aggression and internalizing behavior?...”, and “Page 21. Please explain the following statement…”. Thus, it may be helpful to review our responses to those questions (below) as well for further insight.

Regarding the value of a teacher ratings paper in light of our previous related paper on parent and self-ratings: As we state in the Introduction, teacher ratings, compared to ratings by other informants, of externalizing behavior, and aggressive behavior in particular, offer good prediction of future outcomes (see Verhulst et al. 1994, Whipp et al. 2019). This signals that the observations teachers make are robust enough to predict related future negative outcomes, even when agreement between teachers and other informants is low. Teacher ratings of internalizing problems are not as strong as externalizing behaviors at predicting future internalizing outcomes (see, for example, Sourander et al. 2005), but they are still valuable in future prediction, especially in prediction of the child’s own perceptions of having problems (see, for example, Verhulst et al. 1997). An additional benefit of our study in relation to our previous study (Bartels et al. 2018) is that we can make comparisons to the results from the parent ratings, because the study includes the same cohorts (with many of the same children and large sample sizes). Teacher ratings are much less often investigated regarding co-morbid/co-occurring behaviors, compared to parent ratings, so they are of value in that regard as well. We now make points about these items in the Introduction and Discussion sections.

Regarding teacher’s accuracy in reporting behavior in the school setting: we now discuss this in the Introduction. Peer, teacher and self ratings can all be used in the school setting. Peer nominations are a good choice because they have been shown to predict future outcomes (see, for example, Hämäläinen 1996) and the children can observe their peers inside and outside of class and in and out of sight from adult observers. However, peer nominations do not exist for most of our cohorts. Furthermore, peer–teacher correlations are among the highest of inter-rater correlations (see, for example, Pulkkinen et al. 1999; peer-teacher correlation for behavioral problems was 0.62 while teacher-parent was 0.47 and parent-peer was 0.38). Thus, teachers are a good alternative when peer nominations are unavailable. We now make points about these items in the Introduction and Discussion sections.

Regarding teachers’ ability to assess externalizing and internalizing reliably: please review our extensive response to Reviewer 2 in their 2 questions below: “Do you think teacher reports on internalizing behavior are reliable?...”, “Could the teacher rating explain the moderate correlation between aggression and internalizing behavior?...”. We do acknowledge there are small differences in the reliability of the two scales (see, for example, Polderman et al. 2006; teacher-rated internalizing behavior between MZ twins (intra-twin-pair correlation) were between 0.71 and 0.74, compared to 0.84 for aggressive behavior) regarding teachers as observers, however, we consider them both to be adequately reliable, and make further points about this in the Discussion section.

Regarding the opinions and concerns expressed about teacher perceptions and potential bias: the instruments used in the study (TRF, SDQ, and MPNI) are well-established and well-validated instruments that capture broad psychopathological characteristics. They were created to obtain a perspective outside the clinic regarding children’s behavior. All the items were formulated in such a way that the observer is not making a selection based on opinions or feelings about the child in questions, but about behaviors capable of being directly observed or known about. In this strong foundation (hence, for example, the label ‘check list’), these instruments aim not to be capturing a teacher’s cognitive assumptions or personal criteria but the actual directly observable item asked of the teacher. Regarding potential gender bias, and other potential concerns and considerations in using teacher ratings, we have also added a new Limitations paragraph in the Discussion. 

4) I think the way in which the title is current specified to be quite confusing, as it is mentioned a comparation between 4 different cohorts (and 3 different national contexts). However, this comparative element was not addressed in a structural and empirically proper way. And the author themselves, page 20 in the conclusion section stated, “Although detailed comparison between cohorts regarding behavioral questionnaires, countries and school systems are outside the scope of this paper”. If is that the case, and a structural comparation is challenging given, in the first place, the differences between scales in different cohorts, I would suggest removing the term “comparison” from the title and caution when formulating considerations about similarity in patterns, as it is not in line with what has been statistically tested.

Authors’ response: Thank you for the suggestion to be more careful with our wording in the title. We now suggest a new title: “Teacher-rated aggression and co-occurring behaviors and emotional problems among schoolchildren in four population-based European cohorts”. Additionally, throughout the manuscript we have tried to adjust wording that included “similarity” to show the reader how similar the patterns are without providing the evaluative words. 

5) It is not totally clear to me on which type of aggression the authors are focusing on. In the introduction they provide the distinction between 4 subcategories, namely: proactive, reactive, direct, and indirect. Is the focus of the paper equally on all of them, or just some? Are all these subdimensions caught by the different scales used in the operazionalization? Is there homogeneity in the 4 questionnaires (i.e., are they referring to the same construct)? Concerning this, I think it would be helpful for the readers to provide more directly in the text the different items which were used for computing the aggression measure in each of the 4 questionnaires.

Authors’ response: Thank you encouraging us to clarify these important concerns in our manuscript and in our response here. Reviewer 2 also raised similar questions in 4 different places (our responses to which can also be reviewed): “As mentioned above, you correctly describe the more complex construct of aggression. However, I do not see that you actually assessed all subtypes...” and “I suggest to give more information on what kind of aggressive behavior has been assessed through those questionnaires. Please give item examples.” and “In my understanding conduct problems include more behavioristics than just aggression. Please specify.” and “What types of aggression are the results referring to?”

First, we did not intend to draw specific attention to any particular subtype of aggression, merely to mention that it is a heterogenous behavior. Our 3 questionnaires are well-validated and they capture broad/general aggression, and do not focus on any one particular subtype. Therefore, we have removed or reduced the language of subtypes in the manuscript. In the Methods section, we now give examples of the items collected for aggressive behavior on the questionnaires. In Hendriks et al. 2020, where both the SDQ and CBCL (part of the ASEBA system of questionnaires and similar to the TRF) were given to the same Dutch children, it was found that “genetic correlations indicate that the underlying genetic liability for childhood [aggression] is consistent across measures”. Thus, the SDQ and CBCL get at the same underlying aggression phenotype. Additionally, we have performed a biomarker analysis of aggression (Whipp et al. pre-print on MedRxiv, in press at Scientific Reports) using the MPNI scale in the FT12 cohort and replicated the results in the NTR (using the ASR, the adult self-report questionnaire in the ASEBA questionnaire family). We feel this replication shows that the questionnaires capture a similar aggression phenotype. We include some of these points in the Discussion, lines 1094-1096. 

6) I would provide in the introduction a more clear and complete discussion about existing literature and findings especially concerning the association between aggression and the different co-occurrent behaviors under investigation. I missed especially what are the existing findings, if any, concerning the relation between aggression and prosocial behavior.

Authors’ response: Thank you for this suggestion. We have now re-shaped the Introduction and expanded the literature review in the paragraph about aggressive and co-occurring behaviors. We also get into more detailed aspects of related points in the Discussion section. 

7) I am curious why the author refers to the regression models as “supplemental’ material, as it is supposed to be a central part of the analyses. I would modify this in the text, and probably suggest including the tables in the manuscript rather than as online resource, as this can be the most informative output. Also, I am wondering why the author decided to present both the interaction and the differentiated analyses by gender. I personally find this quite redundant, as the interactions alone should be enough informative.

Authors’ response: Thank you for these suggestions. We have now brought the initial regression models into the main results. We are happy to defer to the editor on their preference for our tables vs. figures to be displayed in the main text vs. put in the Online Resource area. We thought visuals were most helpful for displaying patterns, but of course wished to provide the numbers in tables for scientists to be able to use in the future. 

Regarding gender interactions, we suggest leaving the gender interaction terms in the tables along with presenting the data separately by gender. First, not all cohorts have significant gender interaction terms. Second, interpreting estimates of models with interaction terms is not as straightforward as models that are separated, so providing both allows for perhaps easier interpretation when there is a gender interaction present. Lastly, we thought that, again, if scientists might use this data in the future, for meta-analyses for example, that this information would be easily available. 

Lastly, because Reviewer #2 asked below in ‘Analysis’ “How exactly did you test for gender interaction? What was your model?”, we did clarify the gender interaction text a bit in the Methods section.

Reviewer #2: Thank you very much for this interesting paper. The authors show results on the epidemiology of teacher rated aggressive behavior in school children based on an impressive data set. This paper uses a large, international sample which gives interesting findings on

mean levels of teacher rated aggressive behavior and its associations with internalizing / externalizing behavior and prosocial behavior in 7- to 14-year-old students. It seems that this valuable data has also been analyzed in a longitudinal manner in other papers.

However, bevor publishing I highly suggest to discuss the following major aspects more deeply:

Abstract:

I suggest to explicitly mention the overall aim of this study within the abstract.

I highly recommend to define aggressive behavior within the abstract. As you mention within your introduction there are different types of aggressive behavior. However, the questionnaires you used may assess only parts of the aggression construct.

Authors’ response: Thank you for these suggestions. We have now tried to make our aims more explicit in the abstract (specifically mentioning that we are looking at means, correlations, and gender differences). And since we have made changes to the manuscript in response to both Reviewers questions about aggression subtypes (to remove/reduce this language), we also aimed to not mention aggression subtypes in the abstract. We draw attention in the first sentence of the abstract to aggressive behavior being a broad and heterogenous behavior, and we used the phrase ‘well-defined instruments’ to indicate that the aggressive behavior used in this study is well captured (but explain later in the manuscript about our instruments and how they capture broadly defined aggressive behavior). We hope that this sufficiently addresses your concerns.

Please include information on sample age and school types (if applicable).

Authors’ response: Thank you for this suggestion. We have now added the age range of study participants into the Abstract. However, regarding school types, we are not going to highlight this in the abstract. In the ‘S1 Text’ material we note that Finland and the Netherlands only have publicly funded schools (although there are some ‘private’ schools, they are still publicly subsidized) and even among our UK participants most, but not all, participants are in publicly funded schools. 

Introduction:

As mentioned above, you correctly describe the more complex construct of aggression. However, I do not see that you actually assessed all subtypes. Please specify what kind of aggression you are referring to. In addition, some subtypes may be assessed less valid and reliable within a teacher rating (e.g. relational/ indirect aggression). I suggest to include literature on what kind of aggressive behavior can be reliably assessed though teacher ratings. Also, consider to mention this in your limitation section.

Authors’ response: Thank you for this chance to clarify our text. This question raises similar points as Reviewer 1’s questions #3 and #5, so reviewing our responses to those could provide additional insight beyond our response here. 

In general, we did not intend to draw specific attention to any particular subtype of aggressive behavior, but only to illustrate that aggression is heterogenous. In the revised manuscript, we have now removed or reduced language that focuses on subtypes. The 3 questionnaires used in the study capture more broad/general aggressive behavior, so a focus on subtypes would be inappropriate. Actually, only the MPNI (FT12) has a specific subscale separation for direct and indirect aggressive behavior. Pakaslahti et al. 2000 compared teacher, peer and self adolescent ratings of direct and indirect aggression, and found, for example, correlations between teachers and peers to be the highest (Pulkkinen et al. 1999 also found the same), especially in early and middle adolescence (similar to our study’s age range). Consistency was higher for direct compared to indirect aggression, but no gender differences were found regarding indirect aggression for teacher-peer comparisons. Since peers are a unique observer, able to assess children both in and out of the view of teachers, the fact that their ratings are both reliable and highly correlated with teacher ratings suggests that teachers are well able to assess the two subtypes, despite slight differences in consistency. We now discuss related points in the Discussion, lines 1133-1139.

Page 8, second sentence “Furthermore, children with aggression and comorbidities often have more than one comorbidity and worse prognosis and poorer outcomes …”. Please specify, worse than what? Poorer outcomes with respect to what?

Authors’ response: Thank you for encouraging us to clarify this sentence. We have now modified it to “Furthermore, children with aggression and co-morbidities often have more than one co-morbidity, and, compared to aggressive children without co-morbidities, have worse prognoses and poorer outcomes.” 

Page 8, third sentence (“…regarding the school setting”). Please underlie your statement with literature. What about literature on bullying within schools. In my understanding most studies rely on population-based samples. There are few studies comparing population-based samples with clinical samples (see FemNAT-CD study as one recent example including clinical samples).

Authors’ response: We have now modified the sentence in question to include references and more clarity. We did not want to draw attention to clinical samples since, as mentioned, they are more rare and our study is population-based. We are aware, for example, of Epkins et al. 1993 (compares teacher and self ratings among in-patient and school-based samples) and McConaughy et al. 1994 (discusses teacher ratings in the context of both population-based and clinical samples), however.

Materials and methods

Please specify „handicap or illness that interferes with daily functioning“. How has this been assessed?

Authors’ response: This exclusion criterion was in our original standard operating procedure to the cohorts when we first gathered the data for this study (and for the Bartels et al., 2018 paper). Cohorts have their own variables and protocols for identifying these individuals, but it was meant to capture the more severe disabilities that would limit daily functioning and perhaps keep children out of typical classrooms or be able to behave in typical ways. These disabilities include Down syndrome and severe neurodevelopmental disorders for example. We have now indicated these example conditions in the text to indicate to readers the level of severity we are speaking of. 

Did the excluded sample due to missing values differ to the included sample with respect to any characteristics? E.g., IQ, SES, teacher or child characteristics, internalizing/externalizing behavior?

Authors’ response: In considering this we point out several things. 1) teachers generally complete all/most items on these types of questionnaires (based on our experiences across our large cohorts). And in general, we see a high response rate for the teacher questionnaires in this study (mentioned in the Methods section for each cohort); 2) the GENR, NTR, and TEDS cohorts each allowed missing values in creation of their subscale scores (as mentioned in the Methods section for each questionnaire), only FinnTwin12 did not allow any missing values. Thus, it is likely the most important one to investigate on this issue. 3) Regarding the characteristics mentioned, FinnTwin12 does not collect characteristics on teachers. In the dataset used in the study, we had sex and internalizing/externalizing behavior. We explain in the following question (“Do you have any background information on the samples: IQ, SES, single parent household, psychiatric history? Do the samples differ with respect to IQ, SES, etc.”) about our access to the characteristics such as IQ, SES, and further child characteristics in the cohorts. 4) Using the characteristics we had in the study dataset for FinnTwin12 (sex and internalizing/externalizing behavior), we investigated your question further:

In looking into FinnTwin12, for teacher ratings at age 12, we see that 220 participants (of 4590 teacher rating at age 12 records) had at least one missing MPNI item (and were thus excluded from the study). This is 4.8% of all potential records. In comparing those included in the study and those excluded due to missing, we find no significant differences in sex or internalizing or externalizing scores. For teacher ratings at age 14, we see that 164 participants (of 3013 teacher rating at age 14 records) had at least one missing MPNI item (and were thus excluded from the study). This is 5.4% of all potential records. In comparing those included in the study and those excluded due to missing, we find no significant differences in sex or internalizing or externalizing scores. 

Cohort description:

Do you have any background information on the samples: IQ, SES, single parent household, psychiatric history? Do the samples differ with respect to IQ, SES, etc.

Authors’ response: We appreciate the suggestion to consider these background variables. First, although current analyses do not adjust for these items, since these are all high-income European countries we can roughly assume similarity across cohorts on these items. Also, since these are all population-based cohorts with high response rates, differences from their populations should be minimal. And, even if there are some differences, the associations we found are similar across cohorts. The cohort references in the manuscript do point to recent publications in which some of the basic cohort demographics are described: Kaprio et al. 2013 (FinnTwin12), Kooijman et al. 2007 (GENR), van Beijsterveldt et al. 2013 (NTR), and Haworth et al. 2013 (TEDS). Additionally, for example, Rimfeld et al. (2019) specifically illustrates the representativeness of the TEDS cohort regarding ethnicity and familial socioeconomic status.

While we acknowledge these are useful variables in the understanding of aggressive behavior, being a multi-center cohort study (but cohorts analyzed separately), these variables would further complicate the analysis regarding harmonization efforts and potentially needing to drop participants who do not have these variables available. For example, IQ is only available for a subsample of participants in some of the cohorts. We do have an ACTION paper by Vuoksimaa et al. (2020) using the FinnTwin12, NTR, and TEDS cohorts to investigate aggressive behavior (NTR and FinnTwin12 included teacher ratings) and academic performance that indicated a negative correlation (r range: -0.06 to -0.33) partially explained by shared genetic effects. Furthermore, Stanger et al. (1993) noted that SES only explained 4% of the variance in aggressive behavior as rated with the TRF. These variables would be valuable to look at in future studies to understand their roles as potential mediators/moderators of aggressive behavior and other associations, however, we feel it is out of the scope of this manuscript. 

If I understood correctly you included some children twice in your study for different age groups, as your data stems from longitudinal studies. Do results differ, if you include each child only once? Can you control for this in your model?

Authors’ response: We appreciate the suggestion to consider this. One useful element of keeping our study protocol the same as our Bartels et al. (2018) is for comparability to be maintained, and switching to this approach would reduce that comparability. However, for illustration, we looked at FinnTwin12 (the cohort with the smallest sample size) to investigate this point. If we separate by family ID number (the first half of families in the cohort vs the second half of families in the study) and only include the first half of families in the teacher ratings age 12, and the second half of families in the teacher ratings age 14 analyses, we can compare those mean levels of behaviors to the mean levels of those children in the manuscript. Regarding correlations, differences with those in the manuscript for teacher age 12 ratings were between 0–0.05, and for teacher age 14 ratings between 0.01–0.09. We consider this to be negligible and is unlikely to cause bias in our results. 

It would be highly interesting to use teacher ratings on aggression as a predictor to later aggressive behavior longitudinally.

Authors’ response: We agree this would be interesting. We can point to a few other references that address aspects of this question, however, it is outside of the scope of this paper, so we do not modify our current study design/analyses. 

For example, using the FinnTwin12 cohort, Whipp et al. (2019) showed that both teacher and self ratings of aggressive behavior in adolescence predicted future anti-social personality disorder (of which aggressive behavior is a part). Also using the FinnTwin12 cohort, Vierikko et al. (2006) showed that there is continuity in boys and girls between age 12 and age 14 teacher ratings of aggressive behavior (for boys it was mediated by both genes and common environmental factors, while for girls it was mainly common environmental factors). Using parental ratings in NTR and TEDS, Porsch et al. (2016) showed that stability and heritability of aggressive behavior was high. And Hodgins et al. (2013), in a Canadian cohort, showed that teacher ratings of aggressive behavior at age 6 onward predicted future criminal convictions in early adulthood.

How did you get teacher information? For how long did the teacher know the child? Do you have any information on the teachers and schools?

Authors’ response: This information is currently located in the S1 Text file. We had thought to try to keep the main manuscript Methods section as brief as possible, but if the editor would like us to move some of the information given in S1 Text into the main text, we can do that. As it is, we direct readers to it in the first paragraph in the Methods section lines 671-673 that “Brief cohort and behavioral questionnaire descriptions are presented here (see S1 Text for further details on teacher rating collections and the school systems in Finland, the Netherlands, and the UK).” And again in the Discussion section lines 1085-1089 “Although detailed comparison between cohorts regarding behavioral questionnaires, countries and school systems are outside the scope of this paper (however, see (Hendriks et al., 2019) and S1 Text, for information on these aspects in the cohorts and countries), it is noteworthy that we see similar patterns across cohorts (questionnaires), since the cohorts represent different European countries and the questionnaires were developed for different purposes.”

Study questionnaires

Please consider sharing information on the questionnaires and subscales psychometrics (Conbachs alpha etc.,)

Authors’ response: We have now included a sentence into each of the paragraphs about the study instruments (MPNI, TRF, and SDQ) in the Methods regarding their internal reliability, and have provided references for readers to investigate further.

I suggest to give more information on what kind of aggressive behavior has been assessed through those questionnaires. Please give item examples.

Authors’ response: Thank you for this suggestion. We have now included more information on the types of aggression items in the MPNI, SDQ and TRF in the text in the Methods section, pages 10-12. All the questionnaires collect information on a broad range of aggressive behavior, rather than being narrowed to only one subtype. Additionally, Reviewer #1 also requested something similar in their question #5 above, so you can review our response there as well.

In my understanding conduct problems include more behavioristics than just aggression. Please specify.

Authors’ response: Indeed, ‘conduct problems’ includes both aggressive behaviors and rule-breaking/anti-social behaviors, and the 5 items of the SDQ for ‘conduct problems’ do encompass both aspects. We have noted these two types of items in the text on page 11 (lines 756-757) now, and we had also previously noted (on page 11, lines 754-756) to readers that ‘conduct problems’ was being used as a proxy for aggressive behavior, indicating we were aware that it was not perfectly capturing only aggressive behavior items. One reason for calling the SDQ ‘conduct problems’ as ‘aggressive behavior’ is because it then matches how we categorized ‘aggressive behavior’ in our Bartels et al. 2018 paper (as noted in page 11, line 756), and because in Hendriks et al. 2020, where both the SDQ and CBCL (part of the ASEBA system of questionnaires like the TRF) were given to the same Dutch children, they noted that “genetic correlations indicate that the underlying genetic liability for childhood [aggression] is consistent across measures”. This means that both SDQ’s ‘conduct problems’ and CBCL’s (similar to TRF) ‘aggressive behavior’ are getting at the same underlying aggression phenotype. We reference this in the Discussion, lines 1094-1096.

Analysis

Please consider to control for study site. Especially, as you mention later, that gender differences in internalizing subscale differed between studies. Also, I suggest to control for IQ.

Authors’ response: In the analysis section, we generally had tried to say that ‘each cohort’ was analyzed separately. And we have adjusted the regression modeling paragraph to say (on line 665) “we ran linear regression models for each age and cohort.” We did not combine any cohort results together in the analyses, mainly because the study instrument scales are different, and even though GENR and NTR use the same TRF instrument, we wanted to stay consistent with how we reported the results. 

As for IQ, as mentioned in the above response to the question “Do you have any background information on the samples: IQ, SES, single parent household, psychiatric history? Do the samples differ with respect to IQ, SES, etc.”, not all studies had IQ available for a majority of participants. While we are aware that IQ has been shown to be associated with aggressive behavior (see, for example, Vuoksimaa et al. 2020 ACTION paper), we are unable to include it in this study. 

Did you include aggression items within the overall externalizing problem scale? If so, do I understand correctly, that you include items on aggression both in your dependent and independent variables?

Authors’ response: We have now updated text in our Methods section (last paragraph of Analysis) to clarify this better for readers and avoid confusion. We did not use an ‘overall’ externalizing problem scale in the study or in regression models specifically (for example, the TRF has the ability to combine ‘rule-breaking behavior’ and ‘aggressive behavior’ into the ‘externalizing problems scale’). None of the independent variables used had any aggressive behavior scale questions (dependent variable) as a part of them. We understand that would be problematic if that was the case.

What we meant was that, depending on cohort (and the instrument used), we selected the externalizing problems subscale(s) with the highest correlation with aggressive behavior. So, for TRF we ran two models: rule-breaking behavior (had the highest correlation with aggressive behavior and is officially a part of that instrument’s ‘externalizing problems scale’) and attention problems (although not officially part of the TRF’s ‘externalizing problems scale’, we noted in the Table 1 footnote that we included it since it makes for easier comparison with MPNI and SDQ models). For the SDQ, we used hyperactivity in the model. And, for MPNI, we used hyperactivity-impulsivity in the model. 

How exactly did you test for gender interaction? What was your model?

Authors’ response: Our gender interaction models included aggressive behavior as the dependent variable and another problem scale as the independent variable (e.g., rule-breaking behavior, depression, prosocial behavior) as well as gender and a gender interaction term (i.e., independent variable x gender). We performed a likelihood ratio test using this interaction model and the initial model with only the independent variable and gender included. If the likelihood ratio test p-value was <0.05 (nested models), we reported that there was a gender interaction. Furthermore, our response to Reviewer 1’s question #7 about the gender interaction models may also be of interest to Reviewer 2. 

Results

Please check with the journal specifications: I suggest to include r = and N =, etc.

Authors’ response: Thank you for the recommendation. After consulting the editor’s ‘Journal Requirements’ #1 item’s (above) recommended ‘PLOS ONE's style requirements’ pdfs, and looking at PLOS ONE’s online submission guidelines (https://journals.plos.org/plosone/s/submission-guidelines#loc-style-and-format) in the ‘Statistical Reporting’ section of the ‘Parts of a Submission’ area, we did not locate any formatting specifications for how to report correlations or sample size, etc. It is still possible that we missed something, in which case we would be happy to modify to the journal specifications. We now use ‘r’ in the abstract, but had already been using it in, for example, the Discussion. Regarding ‘N’, we did notice we were inconsistent across Tables regarding use of ‘n’ and ‘N’. We have now cleaned that up. 

Page 15, information on gender differences in internalizing problems compared between studies: I suggest to discuss the different findings. This may be a reason to use site as random effect within the model.

Authors’ response: Thank for encouraging more consideration on this. We actually modified the text in the Results to clarify the results better: “In FT12, girls had slightly higher levels of social anxiety compared to boys (Cohen’s d ranged from -0.21 to -0.23), with boys and girls being more similar for depression (Cohen’s d ranged from -0.09 to -0.12). In GENR, NTR and TEDS, for all internalizing problems, boys and girls generally had similar levels (Cohen’s d ranged from -0.09 to 0.12).” The original FT12 sentence, “In FT12, girls had slightly higher levels of internalizing behavior compared to boys (Cohen’s d ranged from -0.09 to -0.23).”, was combining results too much, creating a mixed message. In fact, social anxiety did show a gender difference, but the depression should be separated from the gender difference message because for age 14 there were no statistically significant gender differences in depression, and the age 12 gender difference was no longer (regarding Cohen’s d) than the other cohorts. Thus, we modified the text. For social anxiety, although we do see a gender difference at the mean level, we do not see notable gender difference in the co-occurrence correlations, which is a common theme of these results: mean level gender differences, but minimal gender difference when it comes to co-occurring behavior. We have now added some sentences to this point in the Discussion, lines 1020-1025.

Regarding controlling for the site/cohort, please see our response above to Reviewer 2’s question “Please consider to control for study site…” 

Discussion

I suggest to discuss more deeply the following aspects:

- What types of aggression are the results referring to? I suggest to look into the following literature: Ackermann, K., Kirchner, M., Bernhard, A. et al. Relational Aggression in Adolescents with Conduct Disorder: Sex Differences and Behavioral Correlates. J Abnorm Child Psychol 47, 1625–1637 (2019). https://doi.org/10.1007/s10802-019-00541-6

Authors’ response: As we note in previous abovementioned queries from reviewers (Reviewer 1 in their #5 question, and Reviewer 2 in their first question in the ‘Introduction’), we have tried to remove or reduce our attention on aggression subtypes. Our aggression measures (TRF, SDQ, MPNI) capture general/broad aggressive behavior, and are not narrowed in to any one subtype of aggression. We have noted that Hendriks et al. 2020 indicated that the SDQ and CBCL (similar to the TRF) capture a similar underlying aggressive behavior, and we have a paper recently accepted at Scientific Reports that suggests a new biomarker of aggression in FinnTwin12 using the MPNI (teacher ratings and others) that was suggestively replicated in the NTR adult cohort using the ASR (adult self-report questionnaire which is part of the ASEBA system of questionnaires, like TRF). We believe these 3 instruments all capture a broad/general measure of aggressive behavior. And, while the MPNI can be broken down into direct and indirect aggression, and the TRF has been utilized to look at violent aggression specifically, and the SDQ has a mix of aggressive behavior and rule-breaking behavior, we do not wish to speculate on aggressive behavior subtypes in this paper. It is outside the current scope. We have tried to now adjust text throughout to make it clear we are aiming to characterize broad/general aggression. And we thank you for pointing us to the interesting reference.

- Do you think teacher reports on internalizing behavior are reliable? May it be more appropriate to use self-ratings for internalizing behavior? Do you have any self or parent ratings with which you could compare results?

Authors’ response: Thank you for encouraging us to consider reliability, Reviewer 1 also asked us to do this in their item #3 (so please review our response there as well). 

Previous studies – including cohorts of the current study – have indicated high reliability for teacher ratings. First, by looking at intrapair correlations in monozygotic (MZ) twins, it is possible to obtain the lower boundary of the reliability (note: the reliability could be lower than this if the trait exhibited low heritability, but aggressive behavior was shown to be between 50–80% heritable in Porsch et al. 2016, using NTR and TEDS). This is based on the assumption that the pair-wise correlation in MZ’s cannot be greater than test-retest correlation in one individual (see e.g., Lykken 1982 for MZ correlations providing information about reliability). For example, Polderman et al. (2006) used the TRF in NTR and showed that intrapair correlations in MZ pairs on teacher-rated internalizing behavior were between 0.71 and 0.74 (compared to, for example, 0.84 for aggressive behavior). Although the intra-pair correlations for internalizing problems were lower than externalizing behavior, the difference was small. Moreover, values of over 0.7 indicate acceptable reliability.

Additionally, Verhulst et al. 1997 indicated that while teacher ratings for internalizing behavior were less reliable than parent or self ratings, they were still valuable and provided unique information in predicting maladjustment (indeed, many studies indicate that teachers and parents provide their own unique and valuable information to studies of childhood psychopathology). 

The focus of our study is on teacher ratings, although we do make some very broad statements about parent ratings comparisons (in the Bartels et al. 2018 publication) in the Discussion paragraph on pages 21, 22, and 25.

- I agree that teachers are a valuable and most likely the most reliable informant on aggressive behavior. However, can you state the following based on your results? (“However, girls were less likely than boys to be identified by teachers as disruptive and referred for treatment “.) Can you say that, although you did not compare with self- and parent ratings? Maybe girls are just less aggressive? Maybe, because relational aggression is less likely to be observed by teachers/ others?

Authors’ response: Thank you for the opportunity to clarify our text. We re-read the sentence in its context and see that the sentence in question is actually referring to the Derks et al. reference in the previous sentence. We see that we may have created confusion for the reader, making it seem like we were speaking about our study findings, when in fact this sentence is still a continued thought from the Derks et al. findings. We have now adjusted the sentence to connect with the previous sentence, making the sentence a little long, but much more clear to the reader. 

- In addition, page 22 (“Teachers are a valuable resource for identifying children in need of specific support and possibly in referral or diagnostic aspects”). I agree, but can you say this along this data? How can you make sure that they capture most children in need of psychological help as you did not compare with a clinical sample?

Authors’ response: Thank you for pointing out that this sentence is off-topic from our results. We have now removed this sentence and inserted this one: “Teachers are a valuable resource for characterizing children’s behaviors, especially externalizing and prosocial behaviors, and we see that although there may be gender differences in separate behavior scales, teacher ratings do not indicate strong gender differences in co-occurring behaviors.”

- Could the teacher rating explain the moderate correlation between aggression and internalizing behavior? As internalizing behavior may not be as visible (cognitions rather than behavior) as externalizing behavior?

Authors’ response: Thank you for encouraging us to consider our interpretation of the internalizing behavior results. It seems that this question is getting at the issue of how reliable teachers are at observing internalizing problems. Our above responses to Reviewer 1’s question #3 and Reviewer 2’s question above starting with “Do you think teacher reports on internalizing behavior are reliable?” could add to this discussion, as well as some additional thoughts here. 

Although teachers are able to characterize externalizing problems very well, they also do well with internalizing problems. Stanger et al. 1993 noted that teachers report fewer internalizing problems than parents, so the variance is lower, which can contribute to the notable disagreement between teacher and parent ratings. However, Verhulst et al. 1994, for example, showed, using the TRF on Dutch children (ages 4-11 years), that teacher ratings on internalizing and externalizing scales were better predictors of poor outcomes than parent ratings. Additionally, Pulkkinen et al. (1999) showed teacher-peer correlations for emotional problem ratings higher (0.34) than parent-peer correlations (0.24). Peers being able to observe children in multiple environments and have conversations outside of adult hearing represent a different type of observer than adults, and peer-teacher correlations being more similar than parent-peer correlations suggests that teachers are able to capture different perspectives than parents. Thus, we believe that while internalizing behavior may not be completely observed by teachers, there is no ideal single informant. We have made some new Discussion section sentences about these points on pages 21-22. 

- You describe aggression dimensional and not on a clinical level. Therefore, are you able to show with your results that higher scores on internalizing behavior may be a „protective factor“ for externalizing behavior? Is there literature to this?

Authors’ response: We only use the word ‘protective’ in the manuscript in the Discussion paragraph about prosocial behavior. The Kokko and Pulkkinen (2000) paper reference longitudinally followed children with teacher ratings of aggression and found that aggressive children with prosocial skills were at lower risk of negative outcomes. 

Regarding internalizing behavior, we don’t want to imply that the lower co-occurrence correlations of aggressive with internalizing behavior (compared to externalizing behavior) suggest internalizing behavior is in some way protective. For example, the August et al. (1996) paper we reference in the manuscript shows that internalizing behaviors are often present with multiple externalizing behaviors. Additionally, for adolescent depression, one of the DSM symptoms is irritability, so in some instances that could exhibit itself as aggression. Likewise, anxieties involving panic or fear could occasionally manifest as aggressive behavior. In FinnTwin12, we only capture ‘social anxiety’, which we do see is slightly negatively associated with aggressive behavior, presumably because being inclined to withdraw from social situations would make it less likely for the individual to engage with others, including aggressively. However, those individuals with social anxiety also have low prosocial behavior (as seen in Table S3), thus, we think this question is quite complex and not able to be accurately addressed with our study design. Willner et al. 2016 seem to suggest that ‘internalizing symptoms may develop as the social consequences of externalizing problems, such as peer rejection, being to take an emotional toll’, so internalizing behaviors do not seem ‘protective’.

- You mention highly interesting results on prosocial behavior and future outcomes. Consider to discuss literature on CU traits to this highly interesting finding.

Authors’ response: Thank you for the suggestion. Since none of the 3 instruments used in the study measure CU traits, we cannot directly speak to their relationship to prosocial or aggressive behavior in these cohorts. But, of course, CU/psychopathic traits often play a role in anti-social personality disorder (ASPD), of which aggressive behavior is also a part (Whipp et al. 2019 in FT12 examined adolescent aggressive behavior and later ASPD for example). And indeed, the DSM-5 includes the specifier ‘limited prosocial emotions’ for conduct disorder (of which 50% of children go on to have ASPD). 

Additionally, we did find some references regarding relationships between aggressive and prosocial behavior and empathy/CU/psychopathic traits and inserted a couple sentences into the Discussion section: “This could be related to issues of empathy (which has been positively linked to prosocial behavior and negatively linked to aggressive behavior [47]) or suggestions to subtype prosocial behavior [48], with studies having shown differences in public/instrumental prosocial behavior being associated with aggression and lack of empathy while more anonymous/non-instrumental prosocial behavior being negatively associated [48, 49].”

- Page 21. Please explain the following statement: “Overall, teacher-based correlations of aggressive behavior with co-occurring behaviors are higher for externalizing problems and lower for internalizing problems and prosocial behaviors than parental ratings at similar ages.“

Authors’ response: Thank you for this suggestion, the paragraph was quite brief, and so now we have added a few sentences to clarify and improve our arguments. This question is also related to Reviewer 1’s questions #3 and #4, so reviewing our responses to those could also be of interest.

---

## [Decision Letter · Decision Letter 1]

31 Mar 2021

Teacher-rated aggression and co-occurring behaviors and emotional problems among schoolchildren in four population-based European cohorts

PONE-D-20-29433R1

Dear Dr.Whipp  

We’re pleased to inform you that your manuscript has been judged scientifically suitable for publication and will be formally accepted for publication once it meets all outstanding technical requirements.

Kind regards,

Gerard Hutchinson, MD

Academic Editor

PLOS ONE

Additional Editor Comments (optional):

Reviewers' comments:

Reviewer's Responses to Questions

**Comments to the Author**

1. If the authors have adequately addressed your comments raised in a previous round of review and you feel that this manuscript is now acceptable for publication, you may indicate that here to bypass the “Comments to the Author” section, enter your conflict of interest statement in the “Confidential to Editor” section, and submit your "Accept" recommendation.

Reviewer #1: All comments have been addressed

2. Is the manuscript technically sound, and do the data support the conclusions?

Reviewer #1: Yes

3. Has the statistical analysis been performed appropriately and rigorously? 

Reviewer #1: Yes

4. Have the authors made all data underlying the findings in their manuscript fully available?

Reviewer #1: Yes

5. Is the manuscript presented in an intelligible fashion and written in standard English?

Reviewer #1: Yes

6. Review Comments to the Author

Reviewer #1: (No Response)

7. PLOS authors have the option to publish the peer review history of their article (what does this mean?). If published, this will include your full peer review and any attached files.

Reviewer #1: No

---

## [Editor Report · Acceptance letter]

16 Apr 2021

PONE-D-20-29433R1 

Teacher-rated aggression and co-occurring behaviors and emotional problems among schoolchildren in four population-based European cohorts 

Dear Dr. Whipp:

I'm pleased to inform you that your manuscript has been deemed suitable for publication in PLOS ONE. Congratulations! Your manuscript is now with our production department. 

Kind regards, 

on behalf of

Dr. Gerard Hutchinson 

Academic Editor

PLOS ONE